# The fat body cortical actin network regulates *Drosophila* inter-organ nutrient trafficking, signaling, and adipose cell size

**Rupali Ugrankar-Banerjee, Son Tran, Jade Bowerman[†, ‡], Anastasiia Kovalenko[§], Blessy Paul, W Mike Henne***

Department of Cell Biology, UT Southwestern Medical Center, Dallas, United States

**\*For correspondence:**
mike.henne@utsouthwestern.edu

**Present address:** [†]Department of Molecular Biology and Genetics, Cornell University, Ithaca, United States; [‡]Department of Molecular Biology and Genetics, Cornell University, Ithaca, United States; [§]Department of Biology, Institute of Biochemistry, Zürich, Switzerland

**Competing interest:** The authors declare that no competing interests exist.

**Abstract** Defective nutrient storage and adipocyte enlargement (hypertrophy) are emerging features of metabolic syndrome and type 2 diabetes. Within adipose tissues, how the cytoskeletal network contributes to adipose cell size, nutrient uptake, fat storage, and signaling remain poorly understood. Utilizing the *Drosophila* larval fat body (FB) as a model adipose tissue, we show that a specific actin isoform—Act5C—forms the cortical actin network necessary to expand adipocyte cell size for biomass storage in development. Additionally, we uncover a non-canonical role for the cortical actin cytoskeleton in inter-organ lipid trafficking. We find Act5C localizes to the FB cell surface and cell-cell boundaries, where it intimately contacts peripheral LDs (pLDs), forming a cortical actin network for cell architectural support. FB-specific loss of Act5C perturbs FB triglyceride (TG) storage and LD morphology, resulting in developmentally delayed larvae that fail to develop into flies. Utilizing temporal RNAi-depletion approaches, we reveal that Act5C is indispensable post-embryogenesis during larval feeding as FB cells expand and store fat. Act5C-deficient FBs fail to grow, leading to lipodystrophic larvae unable to accrue sufficient biomass for complete metamorphosis. In line with this, Act5C-deficient larvae display blunted insulin signaling and reduced feeding. Mechanistically, we also show this diminished signaling correlates with decreased lipophorin (Lpp) lipoprotein-mediated lipid trafficking, and find Act5C is required for Lpp secretion from the FB for lipid transport. Collectively, we propose that the Act5C-dependent cortical actin network of *Drosophila* adipose tissue is required for adipose tissue size-expansion and organismal energy homeostasis in development, and plays an essential role in inter-organ nutrient transport and signaling.

## Editor's evaluation

This important study defines the involvement of the actin cytoskeleton in adipocyte function. The authors present convincing evidence that Actin 5C is a critical mediator of lipid metabolism, nutrient uptake, and larval development in *Drosophila*. This work provides novel insight into lipid metabolism, having broad implications in multiple fields.

## Introduction

Adipocytes are specialized nutrient storage cells that exhibit the remarkable ability to alter their size in response to metabolic challenges such as nutrient excess or development. In mammals, high-nutrient diets can drive the dramatic expansion of adipocyte cell size, termed hypertrophy, that enables them to absorb and store elevated nutrients as fat, thus avoiding lipotoxicity and ensuring metabolic

homeostasis (*Hansson et al., 2019*; *Virtue and Vidal-Puig, 2010*). Dysfunctional hypertrophy is a recognized feature of diseases such as obesity and type 2 diabetes, but the mechanisms governing adipocyte cell size determination and expansion remain poorly understood (*Nunn et al., 2022*).

In *Drosophila*, the central metabolic organ regulating energy storage and signaling is the fat body (FB), functionally analogous to the mammalian adipose tissue and liver (*Arrese and Soulages, 2010*; *Musselman et al., 2013*; *Musselman and Kühnlein, 2018*). Adipogenesis is completed during late embryogenesis when FB cells merge into a single-cell thick continuous tissue comprising of ~2200 cells primed to store nutrients including triglycerides (TG) in lipid droplets (LDs) and excess sugars as glycogen for energy homeostasis (*Bharucha, 2009*; *Hoshizaki et al., 1994*; *Yongmei Xi, 2015*). Upon hatching, young larvae immediately begin feeding and growing; dietary carbohydrates, proteins, and fats are absorbed and metabolized by the gut. Lipids, in the form of diacylglycerol (DAG), are delivered from the gut to other organs including the FB through inter-organ transport (*Heier et al., 2021*; *Palm et al., 2012*). During larval development FB cells do not increase in number, but instead expand enormously in size over a 4–5 day period to accommodate the TG and biomass necessary for metamorphosis (*Meschi and Delanoue, 2021*; *Yongmei Xi, 2015*). Thus, through three instar stages (L1-L3), feeding larvae increase ~200 x in overall biomass, much of which consists of FB LDs (*Sriskanthadevan-Pirahas et al., 2022*; *Jacobs et al., 2020*).

Defects in FB nutrient storage can result in larvae failing to reach their 'critical weight' (CW), defined as the minimum threshold weight at which 50% of larvae can proceed to pupation without further feeding (*De Moed et al., 1999*; *Juarez-Carreño et al., 2021*; *Nijhout and Williams, 1974*; *Tennessen and Thummel, 2011*). This can occur (in part) due to a block in inter-organ lipid trafficking and uptake. Such trafficking is primarily mediated by lipoproteins (denoted as lipophorins in *Drosophila*, Lpp) in the blood-like hemolymph. Circulating Lpp particles receive DAG lipids from the gut, then traverse the hemolymph to deliver them to the FB for LD storage. Lpp particles are themselves synthesized and secreted by FB cells, and loss of FB Lpp production or secretion perturbs inter-organ nutrient flux, causing lipid accumulation in the mid-gut and developmental arrest prior to pupation (*Heier et al., 2021*; *Matsuo et al., 2019*; *Palm et al., 2012*). It has been shown that lipoprotein trafficking to the brain promotes DILP2 secretion from insulin producing cells (IPCs) and influences systemic insulin and nutrient signaling, although the mechanisms underlying this are still being explored (*Brankatschk and Eaton, 2010*; *Brankatschk et al., 2014*). Thus, advancement past the CW checkpoint and successful metamorphosis requires coordination in nutrient uptake, transport, storage, and expenditure during *Drosophila* development (*Merkey et al., 2011*; *Nijhout and Williams, 1974*; *Schönborn et al., 2019*; *Tennessen and Thummel, 2011*). Although studied for over 40 years, a pervasive question in *Drosophila* development is what cellular factors influence larval CW attainment, and how inter-organ communication governs nutrient storage in the FB for CW.

To execute its diverse functions, the FB requires architectural organization. One such architectural feature is the cytoskeleton, composed of protein polymers including F-actin and spectrins that form supportive scaffolds within the FB (*Brozinick et al., 2007*; *Köster and Mayor, 2016*). In fact, the cytoskeletons of both *Drosophila* and mammalian adipose tissues play essential roles in cell homeostasis and nutrient storage, and this relies on poorly characterized mechanisms involving actin dynamics. For example, the actin network of mammalian pre-adipocytes undergoes dramatic remodeling during adipocyte differentiation (*Hansson et al., 2019*; *Liu et al., 2006*). Whereas pre-adipocytes exhibit intracellular actin stress fibers, mature adipocytes accumulate F-actin at the cell surface, forming a cortical actin network together with the spectrin protein fodrin that supports the caveolae-rich surface membrane (*Kanzaki and Pessin, 2002*; *Kim et al., 2019*). Mice fed a high-fat diet also exhibit a significant expansion of adipocyte cell size (i.e. hypertrophy) that requires cytoskeletal rearrangements to harbor fat stores (*Hansson et al., 2019*). How this rearrangement is coordinated remains enigmatic, but enlarged adipocytes exhibit increased F-actin pools and elevated actin modulating factors, suggesting lipid accumulation correlates with actin polymerization. In mice, adipocyte actin networks can regulate the delivery of glucose transporters like Glut4 to cell surfaces for carbohydrate uptake, but whether this is a general regulatory feature of actin networks in adipose tissues remains unclear (*Brozinick et al., 2007*; *Kanzaki and Pessin, 2002*; *Kim et al., 2019*). Thus, a key knowledge gap is understanding how the cortical cytoskeleton of adipose tissues contributes to extracellular nutrient uptake, fat storage, inter-organ lipid trafficking, and adipose tissue expansion, all of which the *Drosophila* FB must coordinate to enable successful development of larvae into flies.

*Drosophila* actins share 98% similarity with humans (*Schroeder et al., 2021*). While humans have as many as 20 actins, *Drosophila* express 6: 2 cytoplasmic isoforms (Act5C and Act42A), 2 larval muscle isoforms (Act57B and Act87E), and 2 adult muscle actins (Act79B and Act88F) (*Burn et al., 1989*; *Röper et al., 2005*; *Wagner et al., 2002*). Despite their close relation, actin isoforms are not functionally redundant, as substituting one for another does not always rescue the phenotype arising from the missing isoform. This is because these actins exhibit highly spatially and temporally regulated expression patterns both in tissues and across organismal development (*Wagner et al., 2002*). Within the FB, F-actin contributes to tissue support and cohesion, but its roles in nutrient and metabolic homeostasis are underexplored.

Here, we utilize the *Drosophila* larval FB as a model adipose tissue to dissect the role of the cortical cytoskeleton in nutrient storage and animal development. We find that the *Drosophila* actin isoform Act5C forms the larval FB cortical actin network, and is required for FB lipid and tissue homeostasis. FB-specific loss of Act5C, but not other actin isoforms, perturbs FB enlargement and larval TG storage. Act5C deficiency leads to small FB cells, resulting in undersized FB tissue, small lipodystrophic larvae, and eventual developmental arrest. Further, we utilize a temporal genetic manipulation approach to reveal that Act5C is required in the expansion of FB cell size during larval feeding and growth. Act5C-loss in the FB also leads to defective organismal nutrient signaling, including blunted DILP2 insulin signaling and reduced feeding behavior. We also find that loss of the actin and spectrin networks differentially impact FB nutrient storage, suggesting these two cortical cytoskeletal systems uniquely influence FB homeostasis. Mechanistically, disruption of the FB cortical actin network does not perturb carbohydrate nor lipid uptake itself, but surprisingly reduces lipophorin (Lpp) secretion from the FB, thus perturbing Lpp-mediated gut:FB and whole body lipid transport. We thus observe a tight correlation between perturbed Lpp secretion and reduced insulin signaling in the FB. Collectively, we identify a non-canonical role for the cortical actin network of *Drosophila* FB cells in regulating inter-organ lipid trafficking and nutrient signaling, and propose that the FB cortical actin network influences adipocyte cell size and tissue growth critical for organismal development.

## Results

### Act5C forms a cortical actin network in the larval fat body

To determine how the actin network is organized within the *Drosophila* larval fat body (FB), we extracted FBs from $w^{1118}$ (laboratory control strain) late feeding L3 larvae, fixed and labeled these with the fluorescent F-actin stain phalloidin, and conducted confocal z-section microscopy. We found that most F-actin was highly polarized to FB cell surfaces exposed to the hemolymph, as well as intra-tissue cell-cell boundaries, with a dim intracellular pool also detectable (*Figure 1A*). Imaging revealed that the cortical F-actin localized to microridge protrusions at the tissue surface, giving the FB surface a 'foamy' appearance consistent with previous reports showing the plasma membrane of FB cells is densely ruffled (*Diaconeasa et al., 2013*; *Ugrankar et al., 2019*, *Figure 1A*).

FB cells contain abundant triglyceride (TG) stores in cytoplasmic lipid droplets (LDs). A subpopulation of small LDs are localized to the cell periphery next to the tissue surface (defined as peripheral LDs, pLDs), whereas larger LDs are packed in the cell interior close to the tissue mid-plane (defined as medial LDs, mLDs) (*Ugrankar et al., 2019*). We examined how this F-actin framework was spatially arranged in relation to pLDs and mLDs. Z-section confocal microscopy of FBs stained with the fluorescent F-actin dye Cell Mask Actin Tracking Stain, together with LD dye monodansylpentane (MDH), revealed a bright ruffled cortical F-actin cytoskeleton that appeared to intimately encircle many pLDs adjacent to the cell surface (*Figure 1B*, **red arrows**). Side-profile *yz-axis* views revealed the actin:pLD arrangement was particularly evident at hemolymph-exposed FB cell surfaces. In contrast, in the tissue interior the actin network did not encircle individual mLDs, but rather localized along cell edges at cell-cell boundaries.

Since the functional relationship between the actin cytoskeleton and LDs in adipose tissues is not well understood, we began to dissect their interplay using larval FBs. First, we utilized a candidate-based visual screen to identify the specific actin isoforms required for the FB cortical actin network. *Drosophila* express six actin isoforms (Act5C, Act42A, Act57B, Act79B, Act87E, and Act88F) in various tissues (*Burn et al., 1989*; *Röper et al., 2005*; *Wagner et al., 2002*). We RNAi depleted each actin isoform specifically in the FB using the FB tissue-specific driver *Dcg-Gal4,* and stained for F-actin.

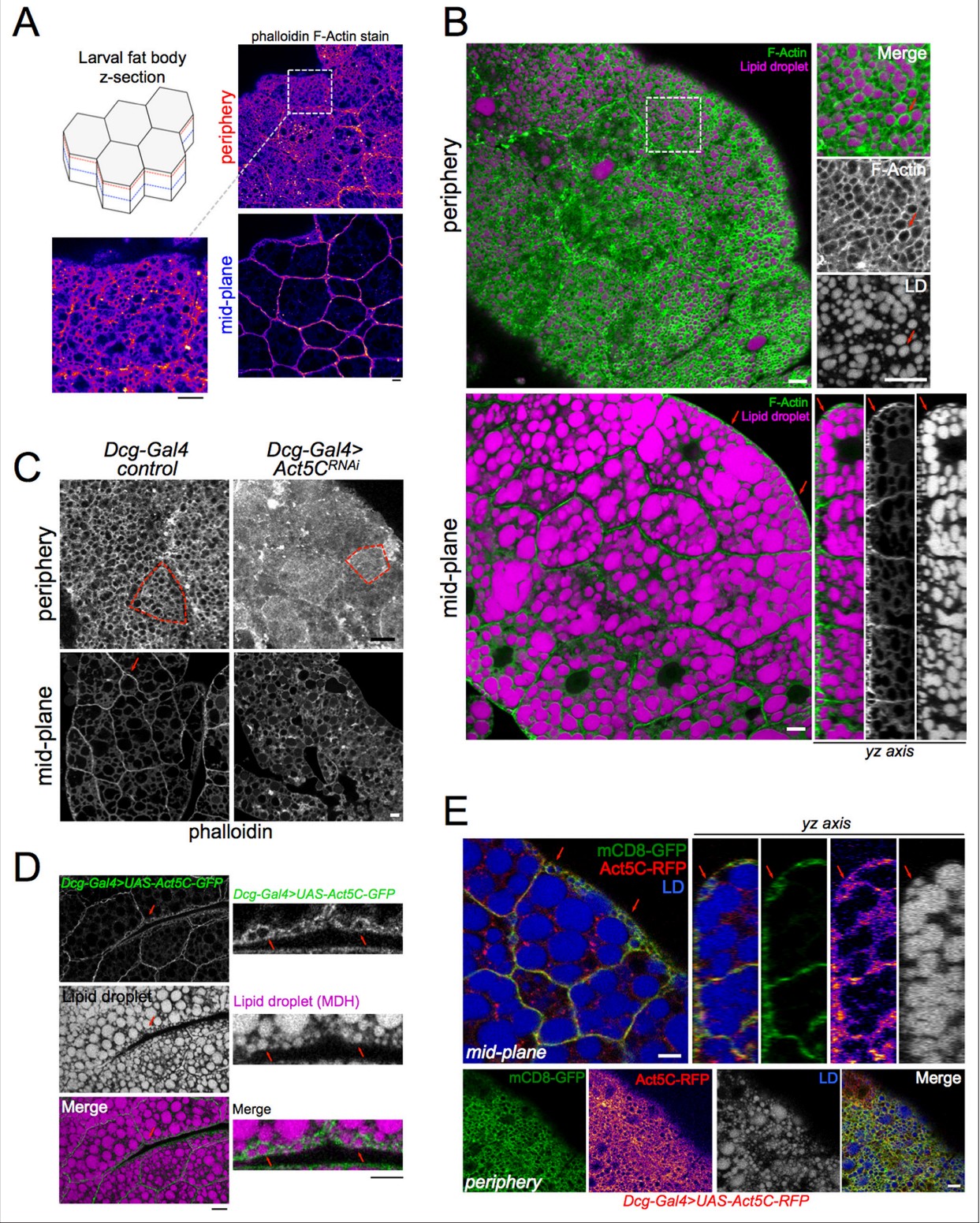

**Figure 1.** Act5C forms a cortical actin network at the surfaces of fat body cells. (**A**) Confocal images of $w^{1118}$ control L3 larval fat bodies (FBs) at their periphery or mid-plane stained with phalloidin, and Fire-LUT pseudo-colored (Fiji). Scale Bar 10 μm. (**B**) Confocal images of control L3 larval FB periphery (*top*) or mid-plane (*bottom*) sections stained with Cell Mask Actin Tracking Stain (green) and LD stain monodansylpentane (MDH, magenta). Red arrows mark peripheral LDs (pLDs) encircled by cortical actin. (*bottom right*) A side yz-axis profile of the z-section of the FB tissue, red arrows marking pLDs encircled by actin. Scale Bar 10 μm. (**C**) Larval FBs stained with phalloidin and viewed from periphery or mid-plane. Scale bar 10 μm.

*Figure 1 continued on next page*

Figure 1 continued

(**D**) Larval FB expressing *Dcg-Gal4 >UAS-Act5C-GFP* and with MDH-stained LDs. Scale Bar 10 µm. (L) Larval FB expressing PM marker mCD8-GFP and Act5C-RFP using *Dcg-Gal4*, and with LD stain MDH. Red arrows indicate pLDs encircled by Act5C-RFP. Scale Bar 10 µm.

The online version of this article includes the following figure supplement(s) for figure 1:

**Figure supplement 1.** Act5C is required for the FB cortical actin network.

Strikingly, targeted RNAi depletion indicated that Act5C (*Dcg-Gal4 >UAS-Act5C^RNAi*), but not any other actin isoforms, dramatically altered F-Actin polarization to cell surfaces and cell-cell boundaries (*Figure 1C*, *Figure 1—figure supplement 1A*). Line-scans confirmed that the major F-actin surface enrichment was ablated in *Act5C^RNAi* FB tissue, indicating the Act5C isoform was necessary for the FB cortical actin network (*Figure 1—figure supplement 1A*). Additionally, the FB cell surface now appeared smooth and lacked its microridge patterning, suggesting Act5C, but not other actin isoforms, was required for this 'foamy' surface appearance (*Figure 1C*, **red boxes are single FB cells**).

To confirm that loss of the FB cortical F-actin network was specifically due to depletion of the Act5C isoform, we utilized quantitative PCR (QPCR) to confirm an ~70% knockdown of Act5C mRNA with our TRiP *UAS-Act5C^RNAi* line (*Figure 1—figure supplement 1B*). We also confirmed that Act5C expression was not significantly affected in the *Act42A^RNAi* line (the other major cytoplasmic actin isoform of the FB). We further confirmed specificity of the *Dcg-Gal4* driver to the FB by examining other tissues, including the gut and brain, extracted from *Dcg-Gal4 >UAS-GFP*-expressing *Drosophila* larvae. We detected GFP signal only in the FB but not in other tissues, validating *Dcg-Gal4* as a FB-specific driver, as previously reported (*Suh et al., 2008*; *Ugrankar et al., 2015*, *Figure 1—figure supplement 1C*). To rule out off-target effects from the Bloomington TRiP *Act5C^RNAi* line, we tested two additional independent *UAS-RNAi* lines for Act5C obtained from the Vienna *Drosophila* Research Center (*Act5C^RNAi VDRC#7139*, *Act5C^RNAi VDRC#101438*) using the *Cg-Gal4* FB driver. Both recapitulated the initial observed *Act5C^RNAi* phenotype, similarly ablating the cortical actin network (*Figure 1—figure supplement 1D*).

To determine whether the TRiP *Act5C^RNAi* line was specifically depleting only Act5C in the FB, and not targeting the other actin isoforms, we co-expressed GFP-tagged versions of all six actins (*UAS-ActXX-GFP*) together with *UAS-Act5C^RNAi* in the FB. Indeed, while we observed that Act5C-GFP normally localized to the cell surface and cell-cell boundaries, the Act5C-GFP protein signal in *Act5C^RNAi* FB was diminished and did not localize to the cell periphery or cell-cell boundaries. Instead, a signal was observed only in the FB cell nuclei. In contrast, the other five GFP-tagged actins, ectopically expressed in *Act5C^RNAi* FB were capable of localizing to the cortical actin network. Thus, none of the other ActXX-GFP proteins appeared to be impacted by *Act5C^RNAi*, indicating that the TRiP *Act5C^RNAi* line specifically targeted only the Act5C isoform (*Figure 1—figure supplement 1D and E*). Indeed, co-staining *Dcg-Gal4 >Act5C^RNAi/Act5C-GFP* FB with phalloidin confirmed that the cortical actin was disrupted in these cells (*Figure 1—figure supplement 1F*). Collectively, this suggests that Act5C is the pertinent isoform necessary for forming the FB cortical actin network, although we cannot rule out that other actin isoforms may play a minor role. These experiments validate our genetic tools, a complete list of which is available in table *Supplementary file 1*.

To further dissect how Act5C forms the FB cortical actin network, we again imaged GFP-tagged Act5C in the FB by expressing *UAS-Act5C-GFP* using the *Dcg-Gal4* driver. Analogous to the phalloidin and Cell Mask Actin staining of wildtype fat tissue, Act5C-GFP decorated FB cell surfaces and enriched at cell-cell interfaces (*Figure 1D*). Act5C-GFP also intimately wrapped around pLDs at the cell periphery (*Figure 1D*, **red arrows**). To understand how this cortical actin network relates to the cell surface, we co-expressed *UAS-Act5C-RFP* together with the plasma membrane (PM) marker *UAS-mCD8-GFP*. This revealed that the Act5C-RFP cortical actin network and PM are closely overlaid, and both exhibit the microridge pattern intimately encircling pLDs (*Figure 1E*, **red arrows**). Collectively, we conclude that Act5C is required for forming the FB cortical actin network, and that it is in close proximity to the cell PM, as well as pLDs adjacent to the cell surface.

## Fat body-specific Act5C loss drastically alters lipid droplet morphology and reduces triglyceride storage

Given that Act5C localizes in close proximity to LDs at the cell surface, we next examined how FB-specific Act5C depletion impacted LDs and fat storage. To quantitatively access this, we measured size changes of the peripheral LDs (pLDs) and mid-plane LDs (mLDs) in larval FBs in which each actin isoform was depleted. Notably loss of Act5C significantly altered both pLD and mLD morphologies and organization (*Figure 2A and B*, *Figure 2—figure supplement 1A and B*). Specifically, Act5C loss significantly increased the average size of pLDs, but drastically reduced the average size of mLDs (*Figure 2A–D*). In contrast, RNAi depletion of the other actin isoforms did not measurably alter the average sizes of either LD pool, although there was a mild increase in average mLD size in *Act57B^RNAi* tissues (*Figure 2D*, *Figure 2—figure supplement 1A*). We confirmed that this LD morphology defect was due to loss of the Act5C isoform, as we observed similar LD disruptions upon Act5C RNAi depletion with the two other VDRC *UAS-RNAi* lines (*Figure 2—figure supplement 1C*). We further confirmed that FB tissue targeted Act5C loss-of-function -resulted in defective LD storage and morphology by driving Act5C knockdown using two other validated FB drivers (*Cg-Gal4*, and *Lpp-Gal4*), both of which caused similar LD perturbations as *Dcg-Gal4* (*Figure 2—figure supplement 1D*, **red arrows**).

To determine how Act5C loss impacted larval triglyceride (TG) storage, we measured TG abundance in *Act5C^RNAi* and control whole larvae. Consistent with their altered LD morphologies, FB-specific Act5C depletion significantly reduced larval TG levels, which were ~10% of 6-day age-matched controls. A significant decrease in total TG was also observed in the two VDRC RNAi lines (*Figure 2E*, *Figure 2—figure supplement 1E*). Since TG levels are influenced by both lipogenesis and TG lipolysis, we employed QPCR to evaluate the expression levels of known lipogenesis and lipolysis factors in isolated FB tissues. Whereas major de novo lipogenesis genes like acetyl-CoA carboxylase (ACC) and fatty acid synthase (FASN) were expressed at normal levels in *Act5C^RNAi* FB compared to controls, genes for TG production including GPAT4, the diacylglycerol acyltransferase (DGAT) enzyme MDY, and the fatty acid desaturase DESAT1 were significantly reduced in expression, suggesting a defect in TG synthesis (*Figure 2—figure supplement 1F*). To determine whether this TG reduction was caused in part by increased lipid mobilization, we also examined mRNA levels of lipolysis-associated genes. The expression of the TG lipolysis enzyme Brummer (BMM), as well as mitochondrial fatty acid oxidation importer CPT1, were significantly reduced, suggesting the observed TG reduction was not a result of elevated lipolysis nor enhanced fatty acid oxidation. Collectively, this suggests that Act5C loss in the FB drastically alters LD morphologies and reduces TG pools, likely due to reduced TG synthesis.

To evaluate how loss of Act5C would influence LDs in a cell autonomous manner, we next utilized the FLP/FRT system to generate chimeric FBs where Act5C was RNAi depleted in only subsets of FB cells (*Konsolaki et al., 1992*). As expected, Act5C-mutant cells from successful mitotic recombination (identified by loss of the RFP marker) displayed a disrupted cortical actin network when stained with phalloidin or Cell Mask compared to adjacent control cells (*Figure 2F and G*, **red arrow**). Notably, Act5C depleted cells also exhibited perturbed pLD morphologies compared to adjacent control cells (*Figure 2G*, **red arrow**). LDs appeared sparser in individual Act5C-depleted cells (similar to general loss of Act5C in the entire FB), and a clear pLD layer was missing in the cell periphery. Some cells instead contained an amorphous fat mass in the periphery near cell-cell boundaries, potentially from the fusion or collapse of the pLD layer (*Figure 2—figure supplement 1G*). This is consistent with the role of the cortical actin cytoskeleton in maintaining LD morphology and organization within FB cells, and suggests that pLDs are encased by the actin meshwork that surrounds them in the cell periphery.

## Fat body-specific Act5C loss leads to small, developmentally delayed larvae that fail to develop into flies

*Drosophila* Act5C global null mutants die as first instar larvae due to arrested growth (*Wagner et al., 2002*), but we noted that FB-specific Act5C depletion produced viable larvae that could proceed through larval development and sometimes pupate. However, we found that age-matched L3 Act5C-deficient larvae were undersized compared to late feeding/pre-wandering controls, suggesting a developmental arrest or significant delay (*Figure 3A*). Consistent with this, control larvae reached critical weight, and commenced wandering and pupating between days ~5–6 after egg laying (AEL),

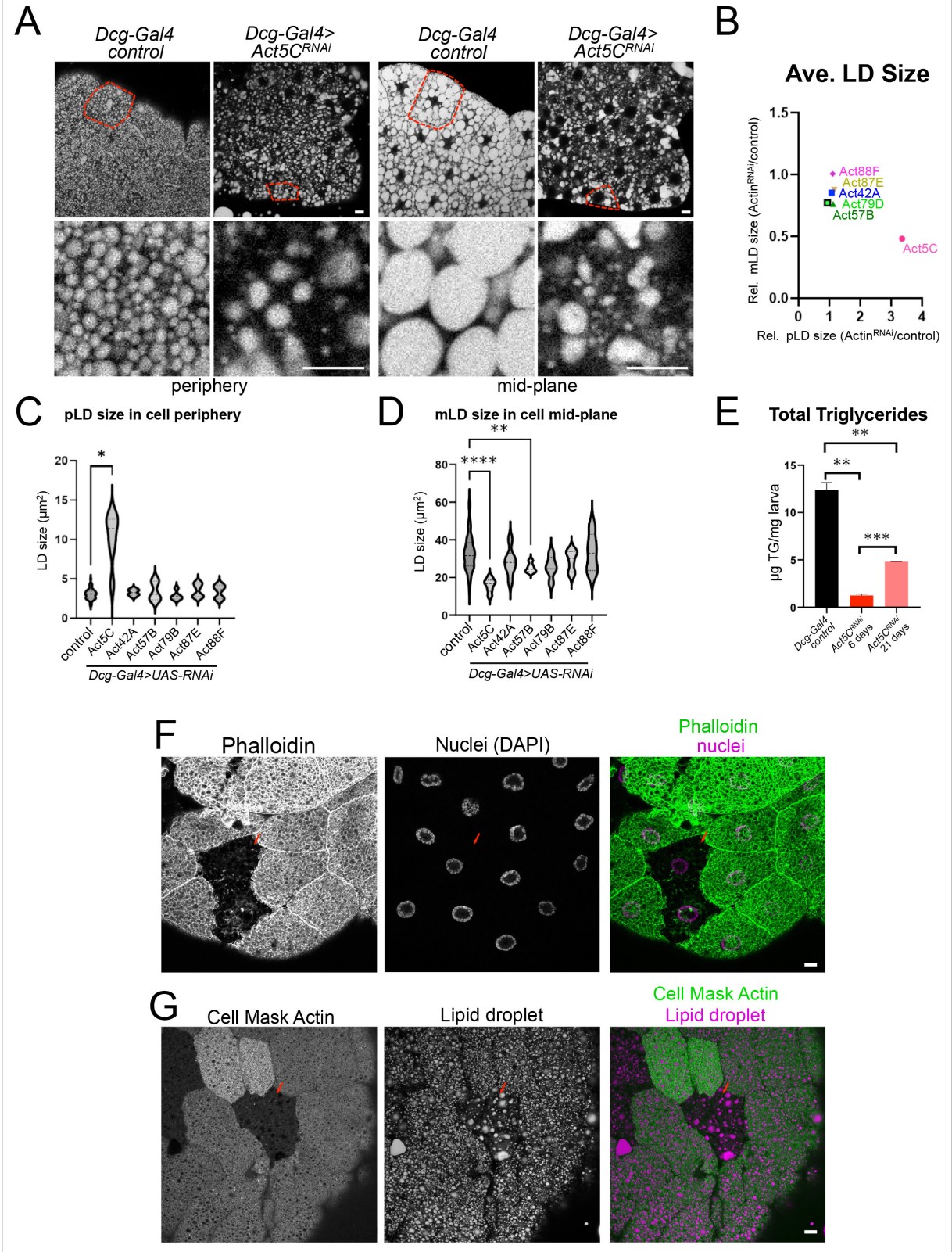

**Figure 2.** Fat body-specific Act5C loss perturbs LDs and reduces triglyceride stores. (**A**) FBs from control (*Dcg-Gal4*) or *Dcg-Gal4 >Act5C^RNAi* larvae stained for LDs (MDH). Scale Bar 10 µm. (**B**) Comparative chart of average pLD and mLD sizes from FB-specific RNAi depletions of six different Actins compared to control LDs. *Act5C* loss significantly altered pLDs and mLDs compared to *Dcg-Gal4* control. (**C**) Average pLD sizes (µm²) for control and FB-specific *Act5C^RNAi* depletions. (**D**) Average mLD sizes (µm²) for control and FB-specific *Act5C^RNAi* depletions. (**E**) Total triglyceride (TG) for age-

*Figure 2 continued on next page*

*Figure 2 continued*

matched control and *Act5C^RNAi* 6 day larvae, and developmentally delayed *Act5C^RNAi* 21-day larvae. (**F**) Confocal sections of larval FBs from FRT/FLP recombination, which enabled loss of Act5C in subsets of FB cells. FBs stained for F-actin with phalloidin (green) and for nuclei with DAPI (magenta). Mutant cells are identified by loss of RFP marker. Scale Bar 10 µm. (**G**) Confocal sections of larval FBs from FRT/FLP recombination, which enabled loss of Act5C in subsets of FB cells. FBs stained for LDs with MDH dye (magenta) and Actin (Cell Mask Actin, green). Red arrows denote cells lacking Act5C, which display defective LD morphology. Scale Bar 10 µm.

The online version of this article includes the following figure supplement(s) for figure 2:

**Figure supplement 1.** Loss of Act5C perturbs FB lipid storage.

whereas there was a significant delay in the growth of Act5C-deficient larvae (*Figure 3B*, and see table *Supplementary file 2* for developmental details). In fact, *Act5C^RNAi* larvae lingered in the food for an additional 7–14 days (~2–3 x slower relative to controls), gradually increasing their overall size (taking up to 3 weeks) to nearly match controls. However, these developmentally delayed larvae appeared transparent, and upon imaging were lipodystrophic with small FBs compared to controls (*Figure 3E*). TG levels of these 3 week old larvae were higher than at day 6, but still much less than L3 pre-wandering controls (*Figure 2E*). A few larger larvae did eventually wander and pupate (<10%), but became necrotic during metamorphosis and failed to eclose to adult flies (*Figure 3B, C and D*, red arrows denote necrotic pupae). Indeed, FBs extracted from wandering *Act5C^RNAi* larvae showed under-developed tissue lobes with significantly smaller cells, suggesting Act5C was required for normal FB development and cell size (*Figure 3F, G and H*). Collectively, this indicates that the Act5C-dependent cortical actin network is required for proper TG storage in the FB, as well as contributes to proper FB tissue and larval growth, and FB cell size.

## Act5C is required in the fat body for post-embryonic larval development

Next, we investigated when Act5C is specifically required in the larval FB, hypothesizing that Act5C may assist in FB cell size expansion during larval development. To enable temporal control over Act5C expression during development we employed the Gal4/Gal80 system (*McGuire et al., 2004*). Gal80 is a temperature sensitive allele that at lower temperatures (18 °C) prevents the Gal4 driver from binding to the UAS (Upstream Activating Sequences), thereby preventing activation of the Act5C RNAi trans-gene, and thus allowing endogenous Act5C expression (Act5C^ON). Upon switching to 29 °C, Gal80 is degraded, allowing Gal4 to bind the UAS and drive FB-specific (*Dcg-Gal4* mediated) knockdown of Act5C, defined here as Act5C^OFF (*Figure 4A*).

First, we allowed *Dcg-Gal4; tub-Gal80ts X UAS-Act5C^RNAi* parents to lay eggs at 29 °C, which were maintained at this non-permissive (Act5C^OFF) temperature through embryogenesis (i.e. 1 day). We then lowered the culturing temperature to the permissive 18 °C for the remainder of larval development to permit Act5C expression (Act5C^ON). Surprisingly, this 1 day Act5C depletion during embryogenesis still enabled larvae to successfully develop into flies (*Figure 4A*), suggesting that expression of Act5C in fat body progenitor cells was not essential for early adipogenesis or FB cell differentiation during late embryogenesis (*Hoshizaki et al., 1994*; *Yongmei Xi, 2015*). However, we cannot rule out that maternally inherited actin may contribute to very early stages of embryo development or tissue morphogenesis (*Burn et al., 1989*).

Next, we extended the Act5C^OFF 29 °C treatment for 2 days and 3 days, depleting Act5C during embryogenesis and through the L1 and/or L2 stages of larval development. Indeed, vials cultured at 29 °C for 2 days still successfully progressed into flies. However, *Act5C^RNAi* cultures maintained at 29 °C for the initial 3 days began displaying varying degrees of arrested larval development and pupal necrosis, although some still progressed into flies, indicating Act5C may be functionally required during the post-L1 stages of larval development (*Figure 4A*). Extending our 29 °C treatment for 4 days resulted in progressively fewer larvae able to develop into flies, and 5 days at 29 °C resulted in the developmental arrest and necrotic pupae observed with constitutive Act5C depletion. Collectively, this indicates that larval expression of Act5C is required for post-embryonic larval development.

To further dissect the developmental time-period during which Act5C is required, we monitored *Act5C^RNAi* larvae hatched and maintained at the permissive (Act5C^ON) 18 °C, then later transferred to the Act5C^OFF 29 °C at different ages. Because 18 °C incubation has been shown to predictably slow embryonic and larval developmental times by a factor of two (*Ashburner et al., 2005*; *Shellenbarger*

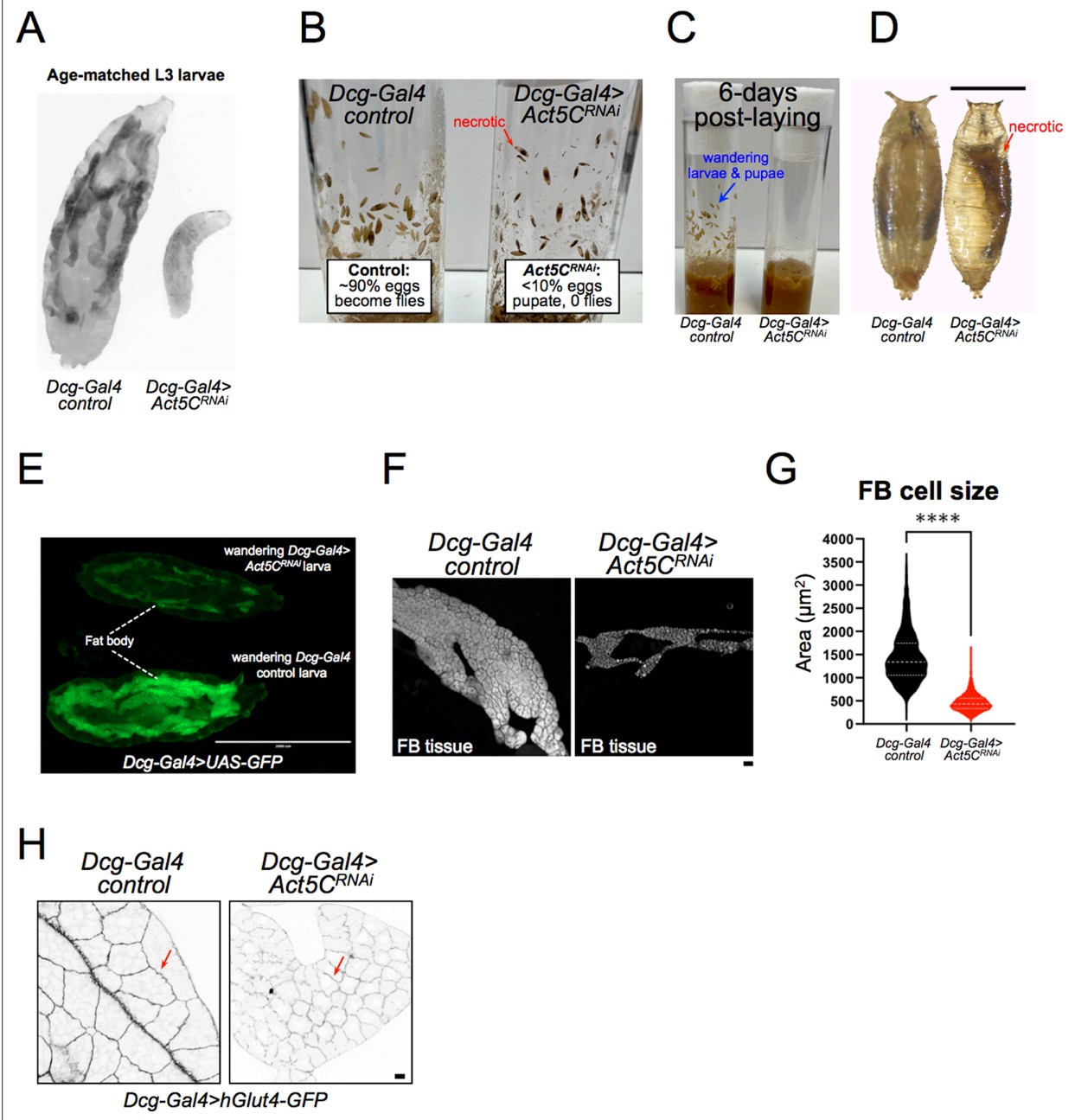

**Figure 3.** Larvae lacking Act5C in fat bodies display developmental and size defects. (**A**) Inverted fluorescent image of age-matched L3 control and *Act5C^RNAi^* larvae expressing UAS-GFP in their FBs. (**B**) Vials with control or *Act5C^RNAi^* pupae. Red arrow denotes necrotic pupae. (**C**) Age-matched control and *Act5C^RNAi^* culture vials, showing L3 wandering larvae and pupae in controls at ~day 6 post-laying, but no wandering of *Act5C^RNAi^* larvae due to developmental delay. (**D**) Comparison of control and *Act5C^RNAi^* maturing pupae. Necrotic pupae marked. Scale bar 1 mm. (**E**) Fluorescence micrograph of L3 wandering larvae expressing soluble GFP in the FB (*Dcg-Gal4 >UAS* GFP). Larvae are either control (bottom) or FB-depleted Act5C (*Dcg-Gal4 >Act5C^RNAi^*, top), latter has a smaller FB and experienced developmental delay prior to wandering (green). Scale bar 2000 μm. (**F**) Fluorescence images of extracted FBs from control or *Act5C^RNAi^* larvae. Scale bar 10 μm. FBs stained with LD stain MDH. (**G**) Average cell size for FBs from control and *Act5C^RNAi^* larvae. (**H**) Representative fluorescent images of larval FBs expressing surface marker hGlut4-GFP to illustrate smaller cell size in *Act5C^RNAi^*. Scale bar 10 μm.

*and Mohler, 1978*; *Yadav and Sharma, 2013*), an adjusted developmental timespan that accounts for this delay was employed. Thus, embryogenesis at 18 °C actually requires 2 'real' days (denoted here as 1 'adjusted development day', or 1 day*). Intriguingly, when cultures were initially placed at the permissive 18 °C for only 0.5 day* or 1 day*, then switched to 29 °C, larvae exhibited developmental

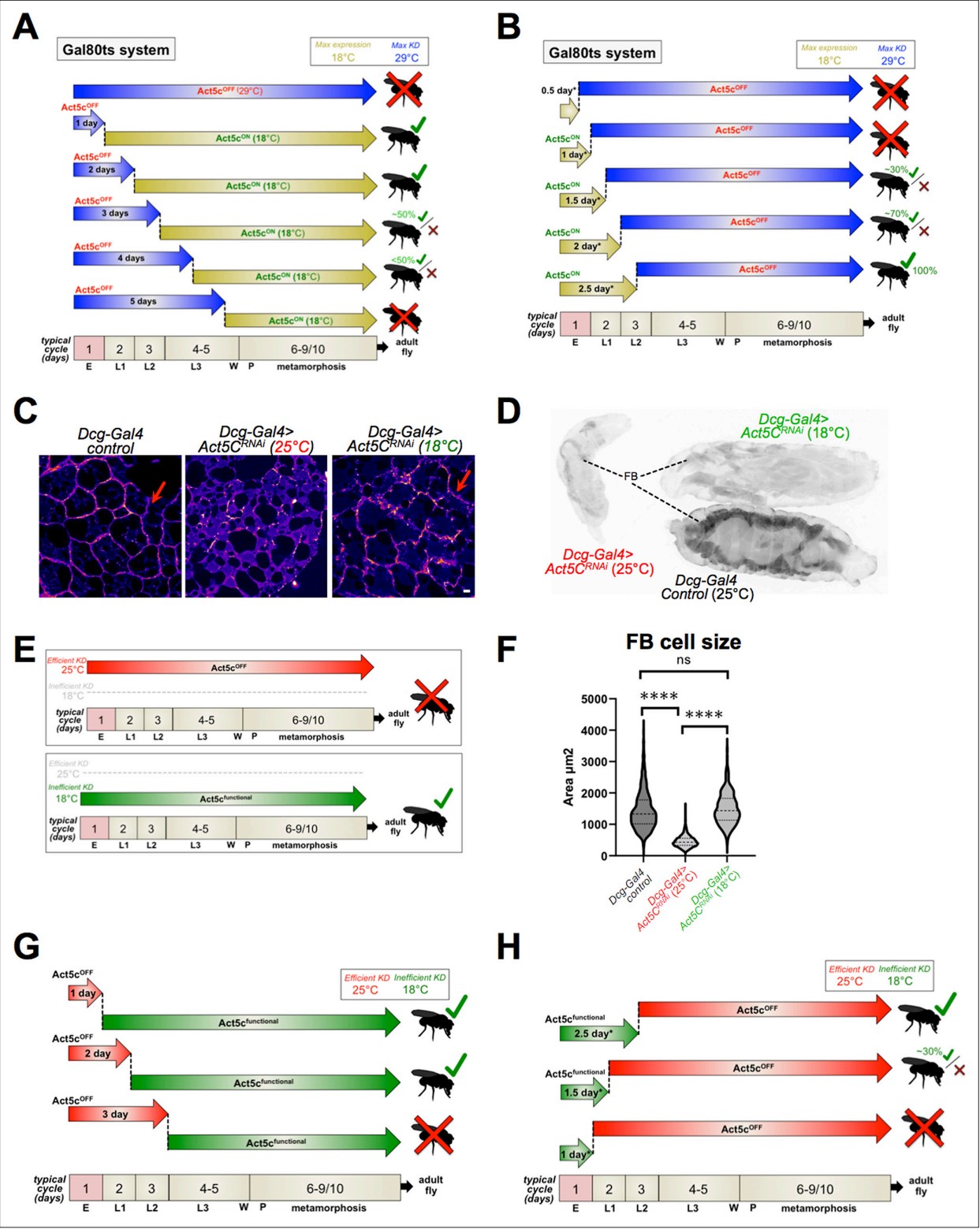

**Figure 4.** Act5C is required post-embryogenesis during larval development. (**A**) Schematic of Gal80 temperature sensitive (Gal80ts) experiments (*Dcg-Gal4;tub-Gal80ts >Act5C*^RNAi^): 29°C to 18°C switch. *Drosophila* egg-laying occurs at 29 °C (maximal Act5C silencing, Act5C^OFF^). Cultures are maintained at this non-permissive temperature for set time-periods, then transitioned to permissive 18 °C to allow endogenous Act5C expression for the rest of development (Act5C^ON^). Their developmental fate (i.e. arrest/necrosis or successful fly eclosion) is indicated. Timeline indicated at bottom. E: embryogenesis, L1, L2, L3: larval instars, W: wandering, P: pupation. (**B**) Schematic of Gal80ts experiments: 18–29°C switch. *Drosophila* egg-laying occurs at 18 °C (endogenous Act5C expressed). Vials are maintained at this permissive temperature for set time-periods, then transitioned to non-permissive

*Figure 4 continued on next page*

Figure 4 continued

29 °C to maximally RNAi-silence Act5C expression for the rest of development. Their developmental fate (i.e. arrest/necrosis or successful fly eclosion) is indicated. Timeline indicated at bottom. E: embryogenesis, L1, L2, L3: larval instars, W: wandering, P: pupation. (**C**) Phalloidin (Fire-LUT)-stained FBs from control, or Act5C$^{RNAi}$ larvae cultured at 25 °C (efficient RNAi-silencing) or 18 °C (inefficient RNAi silencing). Red arrows indicate a cortical actin network at cell-cell boundaries from Act5C expression. Scale bar 10 µm. (**D**) Inverted fluorescence image of age-matched L3 larvae expressing *UAS-GFP* in FBs. Control, Act5C$^{RNAi}$ at 25 °C, and Act5C$^{RNAi}$ at 18 °C are displayed. (**E**) Schematic of time-lapse experiments in *Dcg-Gal4 >Act5C$^{RNAi}$* vials cultured at either 25 °C (efficient Act5C RNAi-silencing, Act5C$^{OFF}$) or 18 °C (some Act5C expressed due to inefficient RNAi-silencing, Act5C$^{functional}$). (**F**) FB cell sizes for L3 *Dcg-Gal4* control and *Dcg-Gal4 >Act5C$^{RNAi}$* larvae cultured at 25 °C or 18 °C. (**G**) Schematic of *Dcg-Gal4 >Act5C$^{RNAi}$* (25°C to 18°C) temperature-shift experiments. Following egg laying, *Drosophila* cultures are maintained at 25 °C (efficient Act5C silencing) for set time-periods, then transitioned to 18 °C to permit low Act5C expression but above a functional threshold. Their developmental fate (as in A and B) is indicated. Timeline indicated at bottom. E: embryogenesis, L1, L2, L3: larval instars, W: wandering, P: pupation. (**H**) Schematic of *Dcg-Gal4 >Act5C$^{RNAi}$* (18°C to 25°C) temperature-shift experiments. Following egg laying, *Drosophila* cultures are maintained at 18 °C (low Act5C expression) for set time-periods, then transitioned to 25 °C for efficient Act5C RNAi-silencing. Their developmental fate (as in A, B, and G) is indicated. Timeline indicated at bottom. E: embryogenesis, L1, L2, L3: larval instars. W: wandering, P: pupation.

arrest and failed to generate flies, suggesting that Act5C expression in the FB *only* during embryogenesis is *not* sufficient for complete *Drosophila* development (*Figure 4B*).

Next, we extended the permissive 18 °C treatment for 1.5 days*, and observed a small, but variable (≤30%) proportion of flies hatching across several culture vials analyzed. Further extension to 2 days* at 18 °C enabled ~70% of pupae to eclose to flies (*Figure 4B*). An almost complete rescue was noted with vials maintained for 2.5 days* at 18 °C. Together, both Gal80ts temperature-shift experiments suggest that larval Act5C is functionally important *after* embryogenesis and during the L1-L3 larval stages of development when the FB tissue is rapidly expanding to reach critical weight (CW). During this developmental period, larvae are actively feeding, growing, and storing gut-derived dietary nutrients in the FB, which requires inter-organ lipid trafficking and signaling to coordinate nutrient storage and FB tissue growth.

As an orthogonal experiment to dissect the developmental time-window when Act5C was required, we took advantage of previous studies showing that simply maintaining *Drosophila* stocks at lower temperatures can limit the degree of RNAi-mediated protein depletion in UAS-RNAi/Gal4 systems, even without the involvement of Gal80 (*Duffy, 2002*; *Perkins et al., 2015*). Indeed, we observed that while FBs of *Dcg-Gal4 >UAS-Act5C$^{RNAi}$* larvae maintained at 25 °C throughout development did not contain a cortical actin network (phalloidin-stained), those maintained at 18 °C exhibited a partial F-actin network, indicating some Act5C was still present at this cooler temperature (*Figure 4C*, **red arrows**). Furthermore, larvae constitutively maintained at 18 °C were generally larger and successfully developed into flies, whereas those maintained at 25 °C exhibited developmental arrest and pupal necrosis, indicating that at 18 °C Act5C was expressing above its functional threshold (*Figure 4D and E*). The Act5C$^{RNAi}$ larvae cultured at 18 °C also displayed improved FB tissue morphology and normal FB cell sizes (*Figure 4D and F*).

Equipped with this additional temperature-sensitive experimental approach, we queried how culturing *Dcg-Gal4 >UAS-Act5C$^{RNAi}$* at 25 °C temperature during egg-laying and early development (Act5C$^{OFF}$), then switching to 18 °C at which RNAi efficiency is reduced (Act5C$^{functional}$), would impact development. Like the Gal80ts system, maintaining cultures for 1 day at 25 °C, then switching to 18 °C permitted full development into flies (*Figure 4G*). Extending the 25 °C timeframe to 2 days still allowed fly development, whereas 3 days at 25 °C led to developmental arrest and necrotic pupae similar to continually maintaining them at this temperature (*Figure 4G*). This further supports a model where Act5C is required later during larval development when the animal is feeding and the FB growing.

Finally, we monitored *Dcg-Gal4 >Act5C$^{RNAi}$* cultures initially reared at the Act5C$^{functional}$ 18 °C, then transitioned to 25 °C. Like the Gal80ts system, cultures kept at 18 °C for only 1 day* (Act5C expressed during embryogenesis only) failed to develop into flies, whereas some larvae from cultures maintained at 18 °C for 1.5 days* could successfully complete metamorphosis into flies (*Figure 4H*). Cultures maintained for the initial 2.5 days* at 18 °C completed normal development into flies, underscoring that expressing Act5C *only* during embryogenesis is *not* sufficient for full development. Collectively, this indicates that larval FB-Act5C expression is required during the post-embryonic feeding phases of larval growth and development when the FB tissue is rapidly expanding and storing nutrients.

## Act5C is required for fat body cell size expansion and fat storage during larval development

Given that larvae are actively feeding and growing during the L1-L3 developmental time window, and that FB-depletion of *Act5C^RNAi* results in developmental delay, shrunken FBs, and small FB cells (*Figure 3*), we hypothesized that Act5C may be required to expand the size of *Drosophila* adipose tissue during feeding and development. Larval FB cells are post-mitotic (*Meschi and Delanoue, 2021*; *Musselman and Kühnlein, 2018*), and therefore FB tissue expansion is accomplished by increasing individual cell size but not cell number. This growth is necessary to accommodate the vast amounts of lipids and biomass for achieving critical weight (CW) and successful progression through metamorphosis (*Merkey et al., 2011*; *Tennessen and Thummel, 2011*). Indeed, our hypothesis is supported by work in mice suggesting that actin reorganization is required to expand adipocyte cell sizes during high-fat diets (*Hansson et al., 2019*). Reinforcing this, our imaging revealed cortical actin networks at cell-cell boundaries within FB tissues, implying intra-tissue actin assemblies may be dedicated to cell ultra-structural support and mechanical cell growth.

To dissect how Act5C dynamics influences FB cell size, we quantified the sizes of FB cells from age-matched L3 control larvae and those with FB-specific depletion of Twinfilin (Twf), an actin effector protein that promotes F-actin disassembly (*Hilton et al., 2018*). Indeed, depletion of Twf in the FB (*Dcg-Gal4 >UAS-Twf^RNAi*) led to significantly decreased FB cell size resembling Act5C loss, indicating that actin dynamics are necessary in the FB for normal FB cell expansion (*Figure 5A*).

Next, we again employed the Gal80ts system to dissect how temporally altering Act5C expression in the FB impacted cell size. We quantified FB cell sizes of age-matched L3 larvae cultured at the permissive 18 °C (Act5C^ON) for the initial 1.5 days*, then moved to 29 °C (non-permissive, Act5C^OFF) for the remainder of development. Of note, this 1.5 day* temperature scheme enables only a few larvae to successfully develop into flies (*Figure 4B*, *Figure 5B*). Indeed, these larvae exhibited significantly smaller FB cells, indicating that loss of Act5C expression after only 1.5 days* suppressed FB cell growth during larval development (*Figure 5B*). Next, we examined FBs from larvae cultured initially at 18 °C for 2 days* or 2.5 days* (followed by Act5C silencing at 29 °C), which in the previous experiments enabled majority of larvae to develop into flies. Indeed, the 2 days* larvae exhibited normal-sized FB cells, and surprisingly the 2.5 days* larvae exhibited slightly larger FB cells than controls (*Figure 5B*). These experiments together indicate that Act5C expression during larval growth is required for normal FB cell enlargement, which appears to correlate with successful development.

To query how these time-specific experiments related to FB TG storage, we also measured larval TG levels of these larvae. As expected, there was a direct correlation between smaller FB cells and reduced TG storage. Larvae from cultures kept at 18 °C for the initial 1 day* or 1.5 days* accumulated significantly less TG than controls. TG levels progressively increased with longer times at the permissive temperature, that is, 2 day* and 2.5 day* (*Figure 5C*). This suggests Act5C is required for FB cell size expansion as well as TG storage during the feeding phases of larval development.

Next, we examined FB cell sizes and TG levels of larvae from cultures initially maintained at the non-permissive 29 °C temperature (Act5C^OFF). Loss of Act5C expression during embryogenesis only (1 day at 29 °C) did not significantly impact FB cell sizes of L3 larvae, and their TG levels mirrored controls (*Figure 5D and E*). This is consistent with their ability to develop into flies. However, extension of this non-permissive 29 °C to 3 days and 5 days caused progressive reductions in both FB cells size and TG stores in these larvae (*Figure 5D and E*). Collectively, this indicates that Act5C is required for FB cell size expansion and TG storage, specifically in the post-embryonic phases of larval development as larvae feed and grow.

## Fat body Act5C loss results in reduced larval feeding and suppressed insulin signaling

The FB is the major nutrient storage organ of *Drosophila*, and controls nutrient signaling pathways for the animal (*Yongmei Xi, 2015*). Since FB-specific Act5C depletion led to smaller FB tissue, reduced TG stores, delayed growth, and developmental arrest, we investigated whether Act5C-deficient larvae displayed altered feeding behavior and/or insulin signaling. L3 control and *Act5C^RNAi* larvae were allowed to feed on a 20% sucrose solution supplemented with Coomassie blue dye. After 1 hr and 4 hr, larvae were examined qualitatively (visually) and quantitatively (by dye extraction and colorimetric measurements) to determine the amount of dye in their gut. Following 1 hr of feeding,<1%

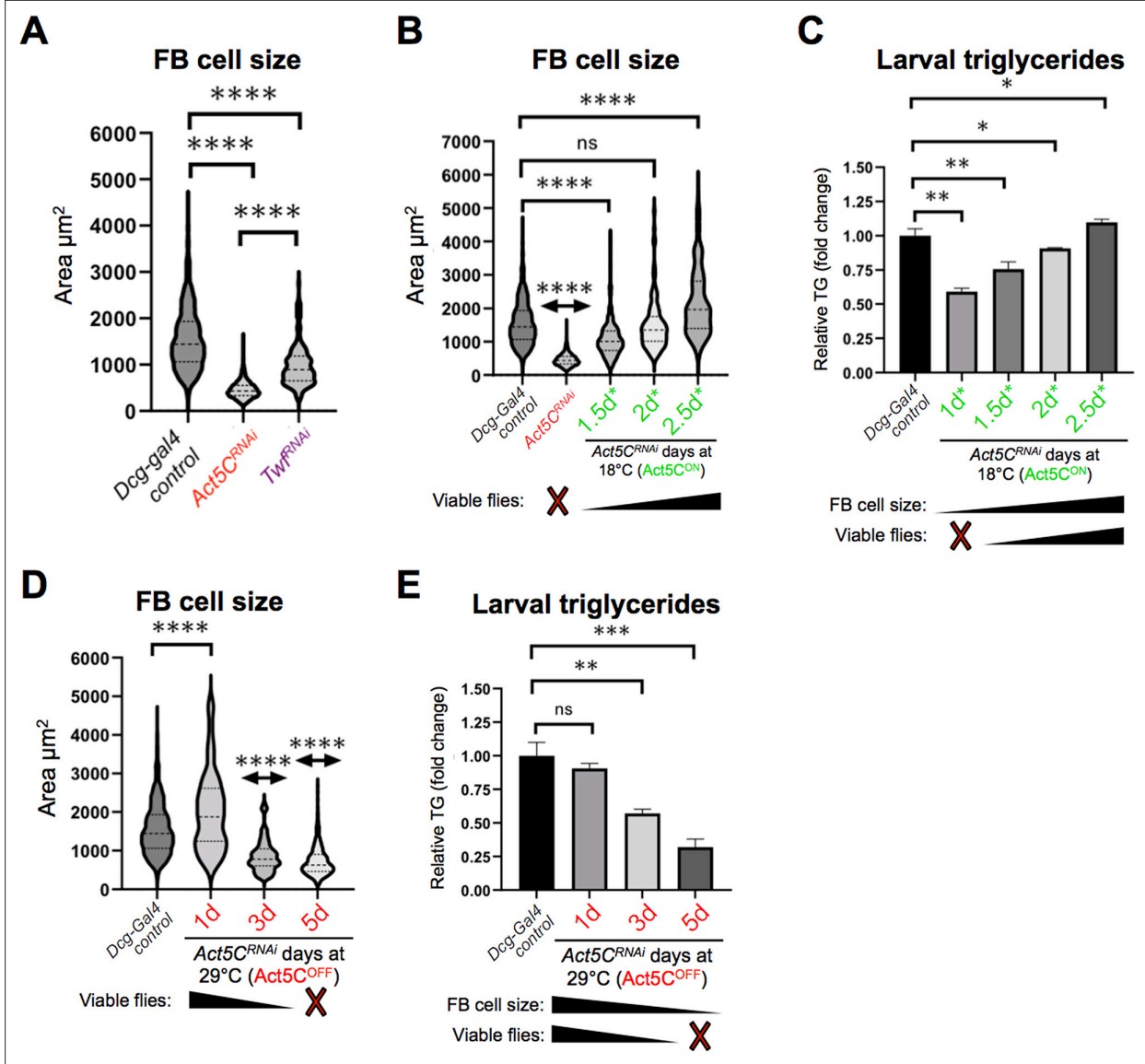

**Figure 5.** Post-embryonic larval Act5C expression is necessary for fat body cell size expansion. (**A**) FB cell sizes for *Dcg-Gal4* control and *Dcg-Gal4 >Act5C^RNAi* or *Twf^RNAi* L3 larvae. (**B**) FB cell sizes for *Dcg-Gal4* control and *Dcg-Gal4;tub-Gal80ts >Act5C^RNAi* larvae from 18°C-to-29°C temperature shift experiments, as in *Figure 6*. The number of days larvae are cultured at the permissive (Act5C^ON) 18 °C before switching to non-permissive 29 °C are denoted. (**C**) Relative changes in TG for control and Gal80ts 18°C-to-29°C temperature shifted larvae (experiments as in *Figure 6*). The number of days larvae are cultured at the permissive (Act5C^ON) 18 °C before switching to non-permissive 29 °C are denoted. How these values correlate to FB cell size and development into flies is also indicated. (**D**) FB cell sizes for *Dcg-Gal4* control and *Dcg-Gal4;tub-Gal80ts >Act5C^RNAi* larvae from 29°C to 18°C temperature shift experiments, as in *Figure 6*. The number of days larvae are kept at the non-permissive (Act5C^OFF) 29 °C before switching to permissive 18 °C are denoted. (**E**) Relative changes in TG for control and Gal80ts 29°C-to-18°C temperature shifted larvae (experiments as in *Figure 6*). The number of days larvae are kept at the non-permissive (Act5C^OFF) 29 °C before switching to permissive 18 °C are denoted. How these values correlate to FB cell size and development into flies is also indicated.

of *Act5C^RNAi* larvae had visible dye in their guts compared to >90% of controls (*Figure 6A*). However, after 4 hr in the dyed food, ~20% to 40% of *Act5C^RNAi* larvae displayed some gut dye, but even amongst those actively feeding, dye amounts were ~50% reduced relative to controls. This suggested a feeding defect in *Act5C^RNAi* animals (*Figure 6A and B*).

As feeding stimulates insulin signaling, we next evaluated how Act5C loss from the FB impacted insulin signaling pathways. QPCR analysis revealed that *Act5C^RNAi* tissues displayed a ~50% reduction in Target-Of-Brain-Insulin (Tobi) transcripts as well as a ~threefold increase in the expression of ImpL2, a DILP2 antagonist, suggesting insulin signaling was blunted in Act5C-deficient FBs

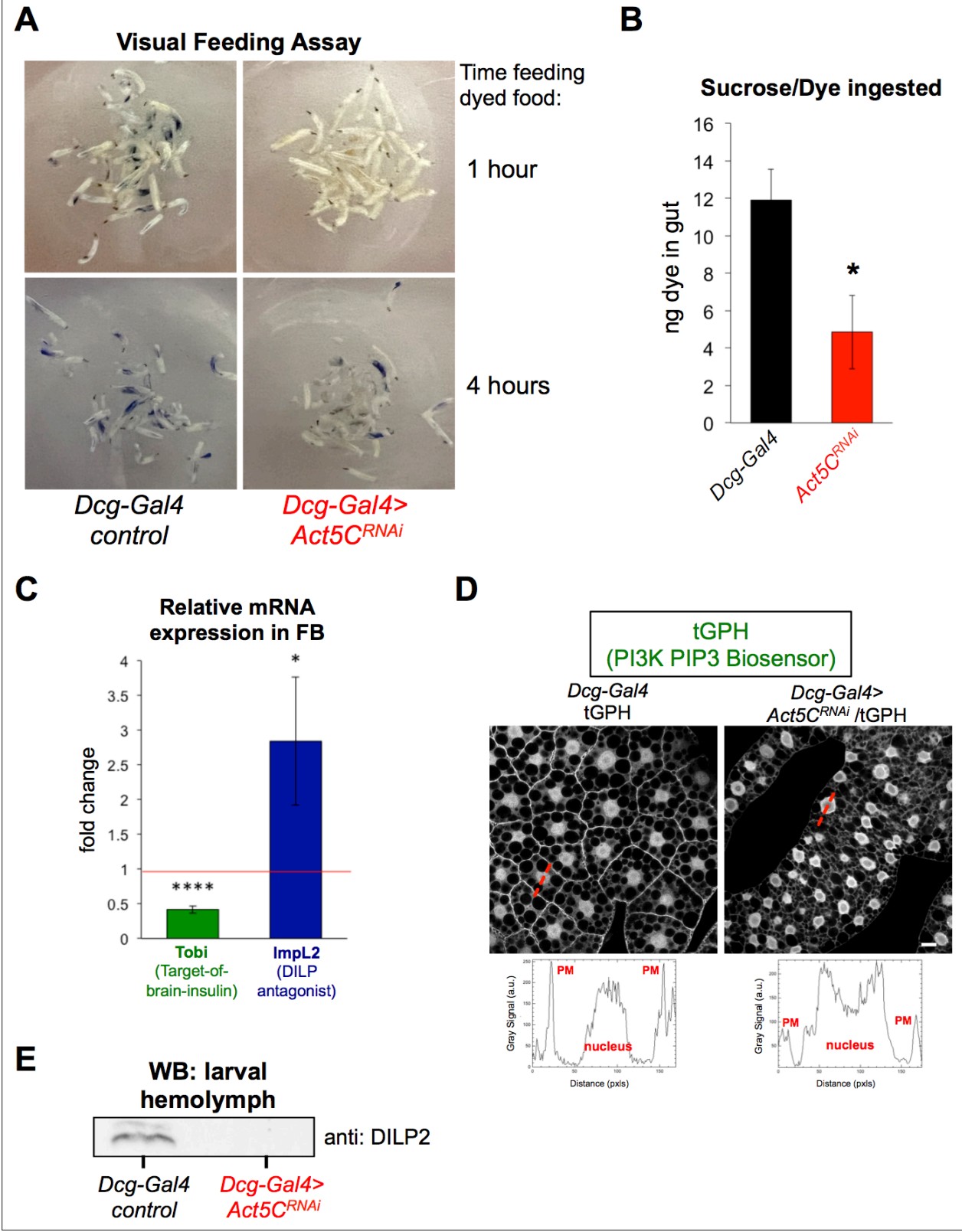

**Figure 6.** Act5C depletion in FB perturbs larval feeding and blunts insulin signaling. (**A**) L3 larvae were allowed to feed on Coomassie blue-supplemented 20% sucrose. Larvae were imaged after 1 hr and 4 hr of feeding. (**B**) Colorimetric quantification of Coomassie blue isolated from larval guts following 4 hr of feeding. *Dcg-Gal4 >Act5C*<sup>RNAi</sup> larvae ingested significantly less Coomassie blue-stained sucrose. (**C**) Relative change in mRNAs for Tobi (Target-of-brain-insulin) and ImpL2 in *Dcg-Gal4 >Act5C*<sup>RNAi</sup> FBs compared to *Dcg-Gal4* control. (**D**) Confocal sections of larval FBs expressing the

*Figure 6 continued on next page*

Figure 6 continued

PIP3-binding fluorescent biosensor tGPH. Linescans with quantified pixel intensities displayed in red with quantifications below. Act5C depletion results in reduced plasma membrane (PM) localization of tGPH. Scale bar 10 μm. (**E**) Western blot using anti:DILP2 antibody of larval hemolymph, loaded for equal volume of hemolymph.

(**Buch and Pankratz, 2009**; **Lee et al., 2021**, **Figure 6C**). We also monitored insulin-stimulated phosphatidylinositol-3,4,5-P3 (PIP3) signaling in FB cells by expressing a GFP-fused PI3K activity reporter (tGPH reporter) (**Britton et al., 2002**). This tGPH reporter contains a PIP3-binding GFP-tagged PH domain ubiquitously expressed under the control of the *Drosophila* tubulin promoter. Re-distribution of tGPH from the cytoplasm to PIP3 at the plasma membrane (PM) is a bioindicator of active insulin-induced PI3K signaling. Indeed, Act5C-deficient FBs displayed reduced PM-localized tGPH relative to cytoplasmic and nuclear signals, indicating reduced insulin signaling in *Act5C^RNAi* larvae (**Figure 6D**). Finally, we directly monitored insulin signaling by extracting larval hemolymph and western blotting for the IPC-secreted insulin-like peptide DILP2 that mediates brain:FB inter-organ insulin signaling. *Act5C^RNAi* larvae displayed significantly reduced DILP2 in their hemolymph compared to controls, again indicating reduced insulin signaling (**Figure 6E**). Collectively, this indicates that larvae lacking Act5C in their FBs display muted feeding and insulin signaling.

## Cortical actin network loss does not reduce the fat body's ability to absorb lipids or carbohydrates

Given that FB Act5C loss perturbed insulin signaling and reduced fat stores, we next focused on mechanistically dissecting how the cortical actin cytoskeleton of adipose cells contributes to nutrient storage. A recent model using mice adipocytes suggests that the F-actin network regulates fat storage by influencing the trafficking of carbohydrate transporters like Glut4 to the cell surface, and thus regulating carbohydrate uptake into adipocytes for eventual storage as TG (**Brozinick et al., 2007**; **Kim et al., 2019**). To determine whether the FB cortical actin network regulated nutrient uptake into FB cells, we developed ex vivo fluorescence-based nutrient uptake assays.

First, we examined whether Act5C loss impacted FB lipid absorption. We exposed extracted larval FB tissues to PBS media containing BODIPY-C12, a fluorescent-conjugated fatty acid that can be internalized by cells and incorporated into TG (**Kasurinen, 1992**). Intracellular BODIPY-C12 could be detected in both control and Act5C-depleted FB tissues (**Figure 7A, B and C**). Surprisingly, the BODIPY-C12 signal in FBs from *Act5C^RNAi* larvae was significantly higher compared to controls, suggesting Act5C loss did not inhibit the capability of FBs to absorb extracellular lipids, and in fact Act5C loss may promote extracellular lipid absorption. Next, we examined whether Act5C loss affected FB carbohydrate absorption. We extracted FB tissues from pre-starved to deplete glycogen stores (**Yamada et al., 2018**), control or Act5C-deficient larvae, and exposed them to PBS media containing 2-NBD-glucose (2-NBDG; **Figure 7D**). 2-NBDG was dimly detected in the cytoplasm of FB cells, and appeared at least as abundant in Act5C-depleted FB tissues as in controls, suggesting Act5C loss did not reduce the ability of FB tissue to absorb extracellular carbohydrates (**Figure 7E and F**).

Carbohydrate uptake and its metabolic conversion to TG help *Drosophila* maintain circulating blood/hemolymph sugar levels and regulate FB TG storage (**Havula et al., 2013**; **Sassu et al., 2012**). Therefore, we also examined whether Act5C loss altered hemolymph carbohydrate levels in vivo. We monitored circulating hemolymph levels of both glucose and trehalose in control and FB-specific *Act5C^RNAi* larvae, and surprisingly found neither was altered (**Figure 7G and H**). Like the ex vivo experiments above, this suggested carbohydrate uptake into the FB was not significantly perturbed with Act5C loss, and rather may even be more efficient; this is intriguing since these larvae have reduced total fat tissue mass, which is the major carbohydrate sink (**Musselman et al., 2013**). The fact that Act5C-deficient larvae exhibited normal circulating carbohydrates was also surprising given that they also manifested reduced insulin signaling (**Figure 6C and D**), suggesting Act5C loss in the FB may have non-canonical impacts on nutrient signaling.

Next, we evaluated FB carbohydrate stores by monitoring tissue glycogen levels. Consistent with the tissue's ability to absorb circulating carbohydrates ex vivo, FB glycogen levels were not significantly different between control and FB-specific *Act5C^RNAi* larvae (**Figure 7I and J**). This underscored that the FB was still operationally capable of absorbing and storing carbohydrates. To further confirm

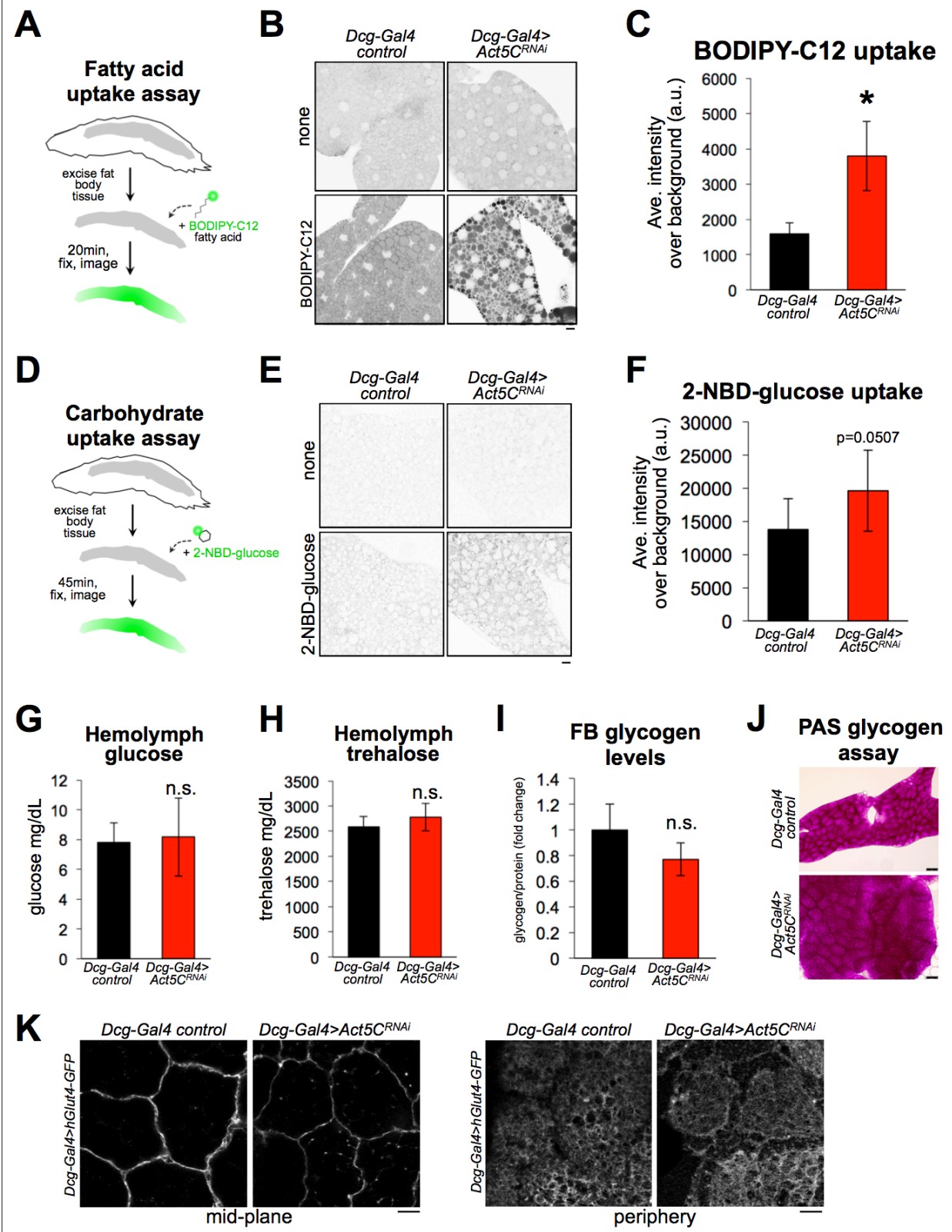

**Figure 7.** Loss of Act5C does not reduce the ability of fat bodies to absorb carbohydrates or lipids. (**A**) Cartoon of BODIPY-C12 fatty acid uptake assays using extracted larval FBs. (**B**) Inverted-LUT fluorescence confocal image of BODIPY-C12-exposed larval *Dcg-Gal4* control and *Dcg-Gal4 >Act5C^{RNAi}* FBs. LD are also stained with MDH. Scale bar 10 µm. (**C**) Quantification of BODIPY-C12 average intensities in larval FBs as in B. (**D**) Cartoon of 2-NBD-glucose (2-NBDG) carbohydrate uptake assays using extracted larval FBs. (**E**) Inverted-LUT fluorescence confocal image of 2-NBDG exposed larval FBs. Scale

*Figure 7 continued on next page*

*Figure 7 continued*

bar 10 µm. (**F**) Quantification of 2-NBDG average intensities in larval FBs as in E. (**G**) Larval hemolymph glucose levels (mg/dL) from control or *Act5C^RNAi^* larvae. (**H**) Larval hemolymph trehalose levels (mg/dL) from control or *Act5C^RNAi^* larvae. (**I**) Isolated larval FB glycogen levels from control or *Act5C^RNAi^* larvae. (**J**) Periodic acid-Schiff (PAS) glycogen stain of larval FBs from control or *Act5C^RNAi^* larvae. (**K**) Confocal images of larval FBs expressing *Dcg-Gal4 >UAS-hGlut4-GFP* at cell edges. Scale bar 10 µm.

that FB tissues could deliver carbohydrate transporters to the cell surface for uptake from circulation, we imaged a GFP-tagged model glucose transporter, hGlut4-GFP (*Crivat et al., 2013*) expressed in larval FBs using the UAS-Gal4 system. As expected, hGlut4-GFP could be successfully trafficked to the surfaces of Act5C-deficient FB cells, suggesting the transport of glucose transporters to the cell surface was not significantly impacted by Act5C loss (*Figure 7K*).

Collectively this indicates that Act5C loss in the FB does not interfere with the capability of the FB to physically absorb extracellular carbohydrate and lipid nutrients, and thus cannot be solely responsible for the reduced TG levels observed in Act5C-deficient FBs.

## Fat body-specific Act5C depletion perturbs lipoprotein-mediated inter-organ lipid trafficking

Since Act5C-depleted FBs appeared capable of absorbing extracellular nutrients, we next examined whether there was an in vivo defect in delivering nutrients to the FB via inter-organ trafficking. The primary mechanism for inter-organ FB lipid delivery is the trafficking of lipophorin (Lpp) lipoprotein-bound DAG via the hemolymph from the animal gut to the FB and other tissues. Lpp particles composed of protein ApoLpp (a homolog of human ApoB) are solely manufactured within FB cells, then secreted into the hemolymph where they travel to the gut, dock, and receive DAG to deliver to other tissues (*Heier and Kühnlein, 2018*; *Matsuo et al., 2019*; *Palm et al., 2012*, *Figure 8A*). A block in Lpp-mediated lipid trafficking leads to lipid accumulation in the mid-gut, which can be visualized as LDs (*Palm et al., 2012*). Since Act5C-deficient FBs displayed reduced TG storage but were capable of nutrient uptake, we next interrogated whether there was a defect in this gut:FB inter-organ lipid transport.

First, we isolated L3 larval guts and stained for LDs using Nile Red. Strikingly, whereas control mid-guts exhibited small sparse LDs, mid-guts from larvae with FB-specific Act5C-depletion displayed significant LD accumulation, and closely resembled larvae lacking ApoLpp (*ApoLpp^RNAi^*), implying a defect in mid-gut lipid export (*Figure 8B*). To determine if other perturbations to F-actin dynamics in the FB could lead to similar defects in gut:FB inter-organ transport, we also RNAi-depleted the F-actin effector Twf in the FB (*Dcg-Gal4 >Twf^RNAi^*). Like Act5C depletion, FB-specific *Twf^RNAi^* loss also resulted in significant fat accumulation in larval mid-guts, indicating that disrupted F-actin turnover in the FB resulted in gut fat accumulation (*Figure 8B*). Indeed, Twf deficiency in the FB resulted in surface actin accumulation in *Twf^RNAi^* FB cells, confirming it perturbed actin dynamics at the FB (*Figure 8—figure supplement 1A*). To quantify relative gut fat accumulation, we isolated guts from these larvae and measured TG levels. Indeed, both Act5C and Twf FB-depletion significantly elevated gut TG compared to controls (*Figure 8C*). As expected, loss of ApoLpp also increased gut TG levels (*Figure 8—figure supplement 1B*).

To further dissect how FB actin perturbation impacted circulating lipids and inter-organ lipid trafficking, we measured larval hemolymph DAG levels, the main lipid in Lpp inter-organ trafficking. Consistent with defective inter-organ trafficking, Act5C loss in the FB significantly reduced hemolymph DAG (*Figure 8D*). Hemolymph DAG pools also trended downward in Twf FB-deficient conditions, but were just below significance threshold due to sample variability. Since hemolymph isolation from ApoLpp-depleted larvae proved challenging, we instead measured whole-larval DAG pools, which were significantly decreased from controls, consistent with the established role of Lpp in trafficking DAG between larval tissues (*Figure 8—figure supplement 1C*). Collectively, this suggests that perturbing F-actin in the FB leads to perturbed gut:FB inter-organ lipid trafficking, reduced circulating DAG, and TG accumulation in the gut.

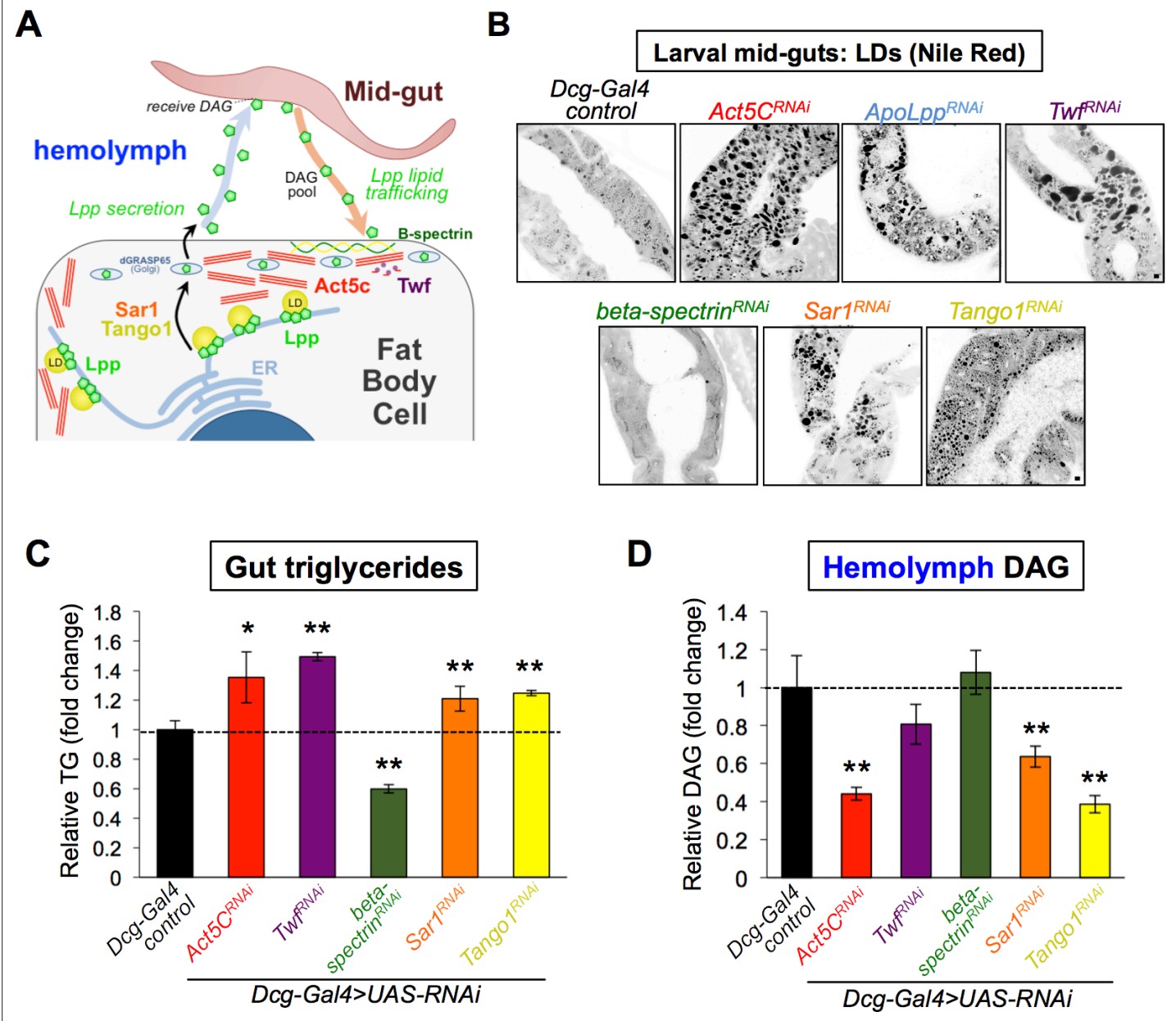

**Figure 8.** Act5C loss alters lipoprotein secretion from the fat body and gut:fat body inter-organ lipid transport. (**A**) Cartoon of lipophorin (Lpp) secretion from FB cells via a Sar1/Tango1-dependent secretory pathway followed by Lpp-mediated mid-gut:FB inter-organ lipid transport,. (**B**) Larval mid-guts from *Dcg-Gal4* control and *Dcg-Gal4 >UAS* RNAi (indicated genes) stained with Nile Red. Scale bar 10 μm. (**C**) Relative TG levels in isolated larval guts from control and *UAS-RNAi* larvae (as in B). (**D**) Relative hemolymph diacylglycerol (DAG) levels in control and *UAS-RNAi* larvae (as in B and C).

The online version of this article includes the following figure supplement(s) for figure 8:

**Figure supplement 1.** Loss of Act5C perturbs FB inter-organ lipid trafficking.

## The fat body cortical spectrin network is dispensable for inter-organ trafficking

In addition to the cortical actin network, FB cells contain a network of spectrin filaments at their surface that provide shaping and structural support for the cell architecture. Since previous work from our lab and others has shown that beta-spectrin localizes to this spectrin network and helps shape the highly-ruffled FB cell surface (*Diaconeasa et al., 2013*; *Ugrankar et al., 2019*), we used this opportunity to also examine how loss of beta-spectrin and its partner alpha-spectrin impacted FB LDs and inter-organ lipid trafficking. Consistent with their role in organizing cell surface architecture that

intimately associates with pLDs, loss of both beta-spectrin and alpha-spectrin altered pLD sizes at the cell periphery (*Figure 8—figure supplement 1D–F*). However, mLDs were not significantly affected, distinct from Act5C and Twf loss (*Figure 8—figure supplement 1F*). In line with this, beta-spectrin loss in the FB did not lead to fat accumulation in the larval gut (in fact, gut TG levels were actually reduced in larvae that lacked FB beta-spectrin) (*Figure 8B and C*). Moreover, hemolymph of beta-spectrin depleted larvae displayed normal circulating DAG levels, indicating that loss of the FB spectrin network did not impact inter-organ lipid trafficking like Act5C loss (*Figure 8D*). Furthermore, we find that F-actin still localized to FB cell-cell boundaries in beta-spectrin RNAi-depleted FBs, although the FB cell surface appeared significantly less ruffled (*Figure 8—figure supplement 1G*). This collectively suggests that spectrin and actin have distinct roles at the FB cell surface; while the spectrin network also helps to shape the highly ruffled surface architecture of the FB, it appears surprisingly dispensable for inter-organ lipid transport.

## Act5C loss in the FB alters the spatial distribution of lipoprotein particles

How does Act5C in the FB contribute to FB:gut inter-organ trafficking? Since hemolymph DAG is bound to Lpp lipoprotein particles that are synthesized and secreted by the FB (*Heier and Kühnlein, 2018*; *Palm et al., 2012*), we hypothesized that Act5C loss impacted Lpp production and/or secretion from FB cells. To interrogate this, we immuno-stained for Lpp particles in larval FB tissue. Notably, in control tissues Lpp particles were concentrated in the FB cell periphery just below the cell surface (*Figure 9A*). In fact, high resolution imaging indicated that Lpp particles formed distinct clusters in close proximity to pLDs adjacent to the cell surface (*Figure 9A*, **red arrows**). This is in agreement with the known spatial distribution of ApoB lipoprotein particles in human hepatocytes. ApoB particles are assembled and lipidated with TG in ER sub-domains immediately adjacent to LDs (*Taghibiglou et al., 2000*; *Ye et al., 2009*). Relatedly, our previous work indicated that depletion of *Drosophila* Lpp particles in larvae impacted pLD homeostasis, suggesting a functional connection between pLDs and Lpp particles at the FB surface (*Ugrankar et al., 2019*).

To further probe the similarities between *Drosophila* Lpp particles and human ApoB, we cultured human Huh7 hepatocytes and immuno-stained for ApoB in cells also co-stained for the ER network and LDs. This confirmed that ApoB forms puncta and crescent-like structures in ER sub-domains adjacent to LDs that appeared very similar to the Lpp puncta we observed in larval FB cells (*Figure 9—figure supplement 1A*). Next, we tested whether Act5C was necessary to maintain this LD:Lpp particle spatial orientation by IF staining Act5C^RNAi FBs for Lpp. In striking contrast to controls, *Act5C^RNAi* FBs displayed a bright Lpp immuno-stain throughout the tissue interior, and the Lpp signal was no longer polarized to the cell periphery near pLDs (*Figure 9A*). The Lpp immuno-signal appeared to localize along a reticulated network that resembled the ER, implying Act5C loss caused some defect in Lpp intra-cellular organization and/or secretion.

To gain higher resolution insights into how Act5C loss impacted Lpp organization and secretion, we conducted thin-section transmission electron microscopy (TEM) on larval FBs. In control tissues, we observed grape-like particle clusters in close proximity to some LDs that resembled Lpp fluorescence immuno-staining (*Figure 9B*, **red arrows**). The clustering of these particles into patches adjacent to LDs also appeared very similar to accumulations of ApoB punctae during their lipidation. By TEM, individual particles were ~60 nm in diameter, in close agreement to the expected sizes of Lpp particles (*Wojczynski et al., 2011*). In further support of this, previous work showing TEM of locust FBs revealed like-sized Lpp particles (*Dantuma et al., 1998*). Moreover, we observed some particles in the microridge folds at the FB cell surface, indicating they may be secreted by FB cells (*Figure 9C*, **blue stars**). To investigate whether these Lpp-like particles were in fact Lpp particles, we did TEM of ApoLpp-depleted FB TEM sections (*Dcg-Gal4 >ApoLpp^RNAi*), and found these particles were no longer observed, supporting that they were Lpp particles (*Figure 9B and C*). A few studies have suggested that similar electron-dense particles may be glycogen particles (*Érdi et al., 2012*), but when we labeled glycogen in the FB using a *UAS-glycogenin-YFP* construct, we found glycogenin had a very different cellular distribution, indicating these LD-associated particles were not glycogen (*Figure 9—figure supplement 1B*).

Strikingly, TEM of *Act5C^RNAi* FBs revealed that the Lpp particles were no longer sequestered in grape-like clusters adjacent to LDs. Instead, these particles now appeared to accumulate inside FB

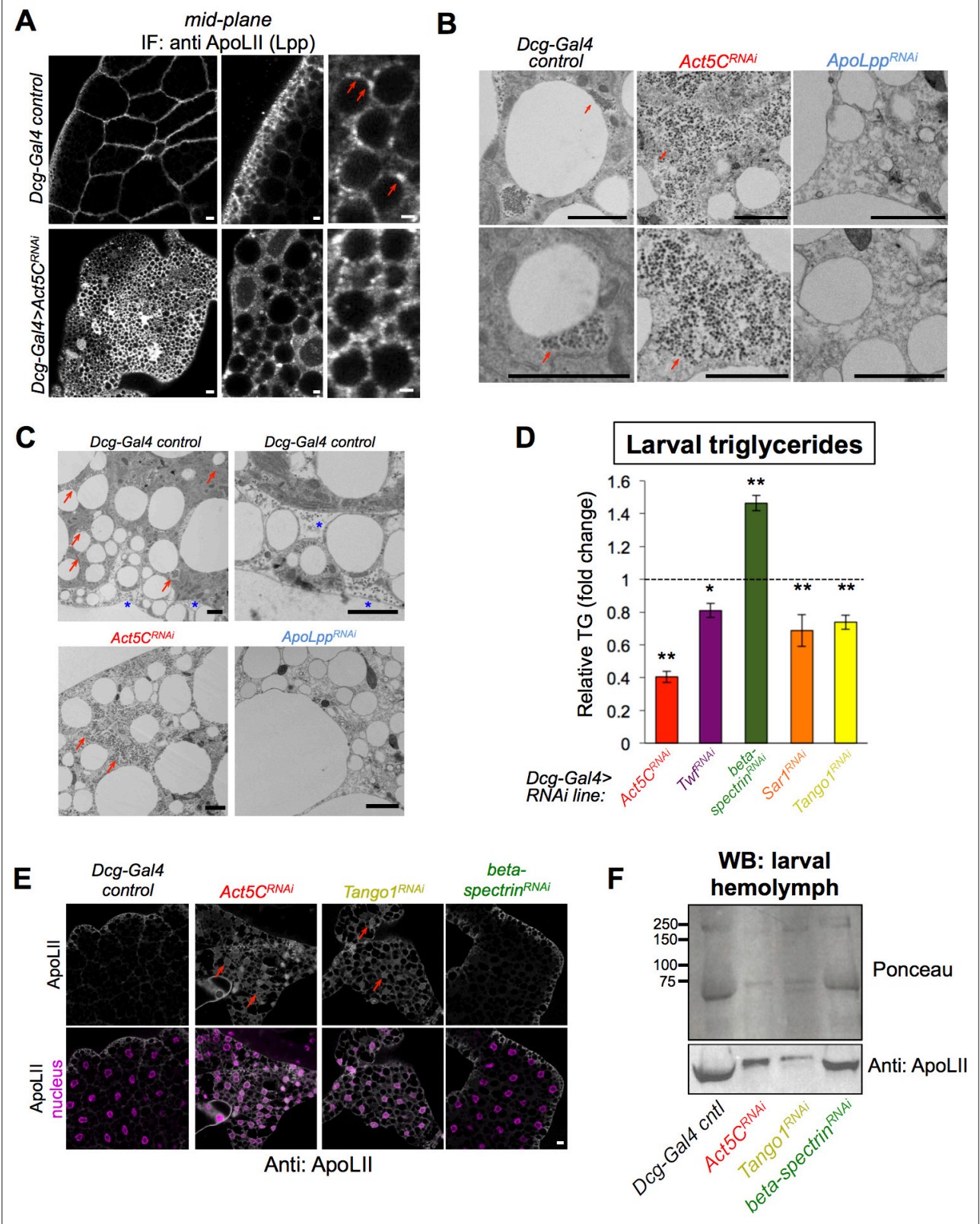

**Figure 9.** Act5C loss perturbs Lpp secretion similar to secretory pathway perturbation. (**A**) Confocal immuno-fluorescence (IF) images of *Dcg-Gal4* control and *Act5C*[RNAi] larval FBs stained for lipophorin (Lpp) lipoprotein component ApoLII (anti:ApoLII) and imaged through tissue mid-plane. Red arrows indicate Lpp punctae adjacent to pLDs (LDs denoted by negative space). Scale bars: (*left*) 10 μm, (*center, right*) 2 μm. (**B**) Transmission electron micrographs (TEM) of FBs from *Dcg-Gal4* control, *Dcg-Gal4 >Act5C*[RNAi], or *Dcg-Gal4 >ApoLpp*[RNAi] larvae. Red arrows indicate Lpp-like particles

*Figure 9 continued on next page*

Figure 9 continued

of ~60 nm diameter, which are absent in *ApoLpp*^*RNAi* FB. Scale bars: 2 µm. (**C**) Negative stain, thin section transmission electron microscopy (TEM) of L3 larvae. FBs are from control, *Act5C*^*RNAi*, or *ApoLpp*^*RNAi* larvae as above. Red arrows indicate Lpp-like particle clusters. Blue stars indicate Lpp-like particles in pockets of extracellular space at the micro-ridges of the FB tissue periphery. Scale bar is 2 µm. (**D**) Relative whole-larval triglyceride (TG) levels for indicated *Dcg-Gal4 >UAS* RNAi lines compared to *Dcg-Gal4* controls. (**E**) Immunofluorescence (IF) micrographs of larval FBs stained for Lpp component ApoLII and DAPI (nuclei). FBs from *Dcg-Gal4* control and indicated *Dcg-Gal4 >UAS* RNAi larvae are shown. Red arrows denote intracellular regions of ApoLII accumulations in RNAi lines, not present in control. Scale bar is 10 µm. (**F**) Ponceau protein stain and Western blot of hemolymph isolated from L3 larvae (equal volumes of hemolymph loaded). The western blot is against Lpp protein ApoLII.

The online version of this article includes the following figure supplement(s) for figure 9:

**Figure supplement 1.** Act5C is necessary for Lpp secretion.

cells and distributed throughout the cell interior, similar to the anti-Lpp IF signal (*Figure 9B and C*). This collectively supports the model that these particles are intracellular Lpp particles prior to their secretion, and that loss of Act5C perturbs their proper spatial organization and eventual secretion out of the FB, leading to defective gut:FB and other (whole-body) inter-organ lipid trafficking.

## Loss of canonical secretory proteins mimics Act5C depletion in the fat body

Since our observations support a model where Act5C is required for FB Lpp secretion, perturbation of this process may explain (at least in part) the reduced LD and TG storage defects observed in Act5C-depleted FB tissues. This may also account for the reduced insulin signaling in Act5C-deficient FBs, as Lpp trafficking to the brain and insulin secretion from brain IPCs have been shown to directly correlate in other studies (*Brankatschk et al., 2014*; *Kelly et al., 2022*).

To further test this model, we compared Act5C depletion to loss of known secretory pathway factors in the FB. We RNAi depleted the known secretory protein Sar1, an Arf family GTPase that promotes ER-to-Golgi trafficking, as well as Tango1, an ER-to-Golgi trafficking factor (*Yang et al., 2021*). Indeed, Tango1 has previously been associated with human ApoB/VLDL secretion (*Santos et al., 2016*). Strikingly, FB-specific loss of either Sar1 or Tango1 and Act5C produced common defects. Both included LD accumulations and TG increases in the larval gut (*Figure 8B and C*). Sar1 and Tango1 loss also mirrored Act5C FB depletion in that they exhibited decreased whole-larval TG levels (*Figure 9D*), as well as lower hemolymph DAG pools (*Figure 8D*). Tango1-depleted larvae also developed L3 developmental arrest, consistent with a block in Lpp secretion and inter-organ lipid transport (table *Supplementary file 2*). In line with this, staining for Lpp particles in Tango1-depleted FBs revealed an identical build-up of intracellular Lpp in FB cells as Act5C loss, indicating that loss of Act5C or Tango1 led to similar defects in Lpp secretion (*Figure 9E*, **red arrows**). As an additional interrogation of how Act5C loss impacted the FB cell secretory pathway, we imaged *UAS-dGRASP65-GFP*, a known Golgi marker protein. As expected, dGRASP65-GFP marked numerous Golgi foci in control FB tissues, particularly in the cell periphery just below the surface (*Figure 9—figure supplement 1C*). However, dGRASP65-GFP foci were reduced in abundance in Act5C-depleted FBs, particularly in the mid-plane, indicating Act5C loss impacted Golgi homeostasis in the FB.

To quantitatively access how Act5C loss impacted Lpp and general protein secretion, we isolated larval hemolymph and western blotted for circulating Lpp protein ApoLII. Indeed, RNAi depletion of either Act5C or Tango1 led to reductions in secreted ApoLII in the hemolymph, consistent with a block in Lpp release from the FB (*Figure 9F*). Similarly, general protein staining (Ponceau stain) also revealed reduced secreted proteins in *Act5C*^*RNAi* and *Tango1*^*RNAi* hemolymph, indicating loss of Act5C possibly impacted secretion of other proteins and/or adipokines from the FB. In contrast, beta-spectrin-depleted FBs showed no major defects in ApoLII and protein secretion.

Collectively, this suggests that Act5C and the cortical actin network are required for normal Lpp particle secretion into the larval hemolymph, subsequent lipid release from the gut and Lpp-mediated lipid transport required for normal TG storage in the FB. In line with this, both ApoLpp and Act5C FB depletion produce analogous phenotypes such as reductions in larval FB cell size, and developmental arrest (*Figure 9—figure supplement 1D*, table *Supplementary file 2*). Previous work has also indicated that defects in Lpp trafficking are closely correlated with reduced insulin signaling in *Drosophila*, indicating functional crosstalk between lipoprotein and insulin signaling (*Brankatschk*

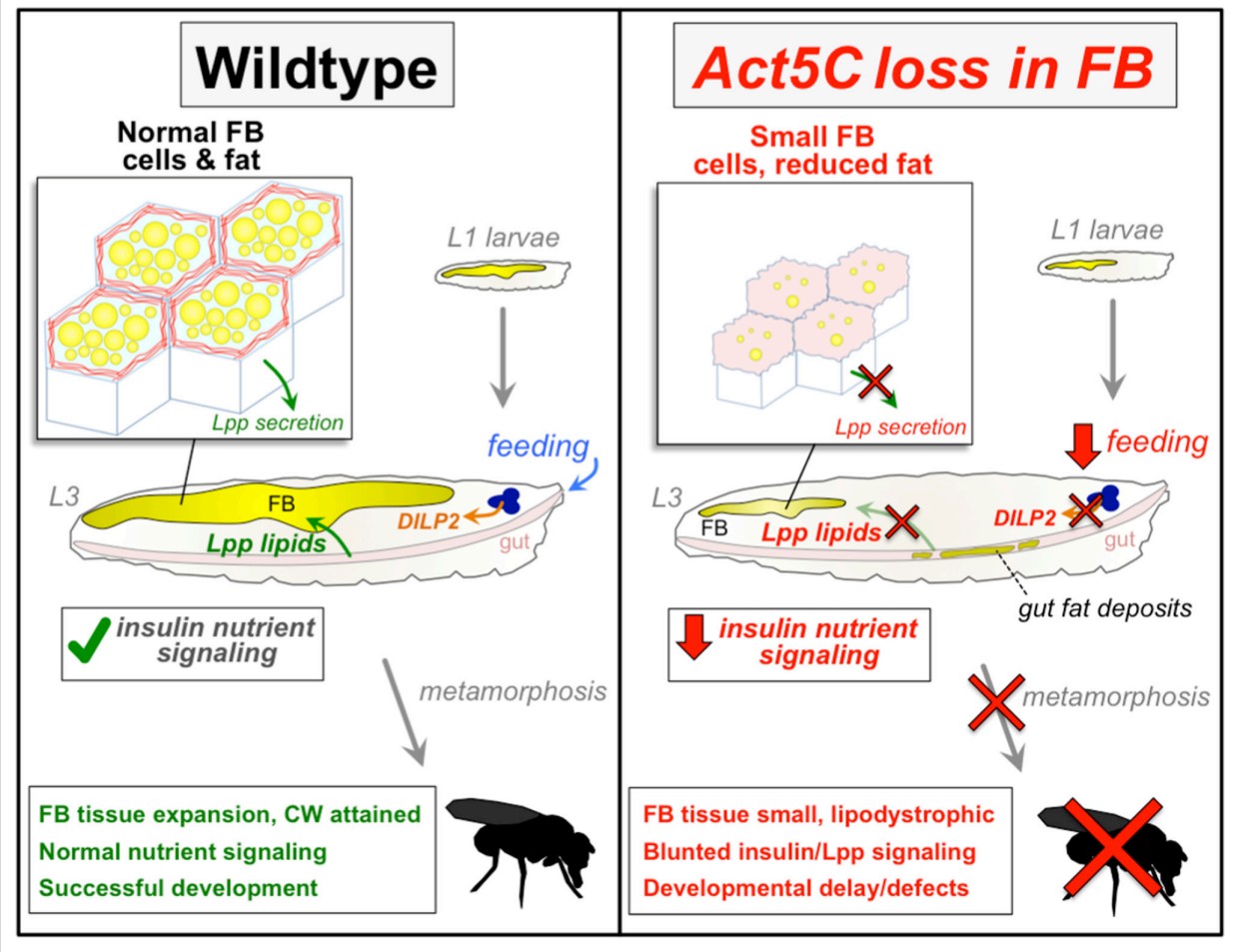

**Figure 10.** Working model for role of Act5C in FB cell and tissue expansion, Lpp-mediated inter-organ lipid transport, nutrient signaling, and fly development.

*et al., 2014*). Thus, Act5C appears to play an essential role in Lpp-dependent inter-organ lipid trafficking and FB development in *Drosophila* that ultimately influences nutrient signaling and storage (*Figure 10*).

## Final conclusions and limitations of study

Altogether, our work suggests that Act5C is necessary for larval FB cell and tissue expansion during larval development, as well as normal Lpp-mediated lipid trafficking and insulin signaling (*Figure 10*). Loss of Act5C or defects in F-actin turnover in the FB (*Twf^RNAi*) lead to disrupted Lpp secretion into the hemolymph, and consequently reduced inter-organ lipid transport, gut fat accumulation, and lower FB fat storage. This Lpp secretion defect mimics loss of canonical secretory pathway factors like Tango1 or Sar1. Additionally, we find that FB Act5C depletion leads to reduced larval feeding and suppressed insulin signaling. The mechanism underlying these signaling changes is not fully understood, but it is possible that these are primarily due to disrupted nutrient signaling. Past work has shown that Lpp particles accumulate at neurons that harbor insulin-like peptides and influence their release (*Brankatschk and Eaton, 2010*). More recent studies reveal that high sugar *Drosophila* feeding promotes Lpp-mediated crosstalk with the brain and ultimately influences insulin signaling and feeding behavior (*Kelly et al., 2022*). A key emerging question is how Lpp lipid trafficking influences inter-organ insulin signaling and feeding, and how this impacts organismal homeostasis and development. Future studies will dissect how Act5C, FB adipose tissue growth, and Lpp signaling regulate nutrient signaling and feeding.

## Discussion

The actin cytoskeleton is an essential scaffold for cell integrity and plays roles in organelle homeostasis as well as organismal growth and development (*Hansson et al., 2019*; *Ojelade et al., 2013*). In mammals, adipocyte differentiation and size expansion require actin network remodeling, and defects in actin-associated processes are associated with metabolic diseases like type 2 diabetes (*Hansson et al., 2019*; *Kim et al., 2019*). Clear knowledge gaps remain regarding how actin influences nutrient uptake, storage, adipocyte cell size, and insulin signaling. Here, we dissected how the cortical actin network of *Drosophila* larval FB cells contributes to biomass accruement during larval development, thereby influencing nutrient storage and signaling. In essence, we find that Act5C plays an essential role in post-embryonic larval growth of adipose tissue, as well as inter-organ nutrient trafficking. Moreover, we mechanistically reveal that Act5C is required for the cortical actin meshwork that shapes the FB cell surface and cell-cell boundaries. Surprisingly, loss of this actin network does not reduce the capability of FB cells to absorb carbohydrates nor lipids, but rather perturbs the FB release of Lpp particles, which mediate lipid trafficking from the gut to the FB and other organs. Defects in this Lpp release likely correlate with reduced insulin signaling. We also find the cortical actin meshwork is necessary to organize the pLDs residing adjacent to the cell surface, as acute Act5C depletion targeted in whole FB tissue as well as in single FB cells (generated via FLP/FRT methodologies) results in loss of the pLD layer. We also find that loss of Act5C and cytoskeletal dynamics specifically during larval developmental stages of rapid feeding/growth interferes with the ability of FB cells to expand to accommodate nutrients, leading to undersized FB cells and consequently reduced lipid stores. This results in lipodystrophic larvae that fail to reach 'critical weight', are developmentally delayed, and unable to complete metamorphosis.

A major question in adipose tissue biology is how adipocytes regulate their size to enable fat storage. We propose that the Act5C pool at cell-cell boundaries contributes to maintaining tissue architecture as well as promoting cell size expansion during larval feeding and growth. Such cell size expansion is necessary to provide FB tissue with sufficient volume for TG storage and biomass accumulation necessary for achieving larval 'critical weight' (CW). In support of this, FB-specific Act5C depletion yielded undersized FB cells with smaller and misshapen mLDs. Consequently, larvae suffered from generalized lipodystrophy, reduced TG storage, and smaller body size relative to age-matched controls. Few Act5C-deficient larvae reached CW for pupation even after a significant delay of 7–14 days. Moreover, *Act5C^RNAi* larvae that did pupate became necrotic, presumably because they lacked the energy and/or biomass for adult tissue differentiation. Similar lipodystrophic *Drosophila* models, such as larvae lacking the lipin homolog dLipin, also exhibited undersized FBs containing small LDs, developmental delay, and pharate adult lethality (*Ugrankar et al., 2011*). Thus, the failure of *Act5C^RNAi* larvae to complete development suggests that appropriate FB lipid storage and biomass increase during larval feeding stages requires an intact FB cortical actin cytoskeleton.

Larval development can arrest for several reasons, including poor nutrient availability, defective nutrient storage/utilization, or inefficient delivery to tissues (*Tennessen and Thummel, 2011*). A block in Lpp-mediated inter-organ lipid trafficking likely also contributes to the observed developmental arrest in Act5C-deficient larvae. How does Act5C mechanistically contribute to Lpp secretion and gut:FB nutrient trafficking? We find that the FB actin cytoskeleton closely associates with surface membrane microridges and pLDs, suggesting Act5C influences lipid homeostasis and potentially lipid exchange between the hemolymph and FB cells at the cell surface. In line with this, immuno-staining and TEM reveal Lpp particles cluster adjacent to pLDs, consistent with reports indicating that ApoB-containing lipoproteins and LDs are lipidated with TG at common ER sub-domains (*Taghibiglou et al., 2000*; *Ye et al., 2009*). Indeed, our previous work revealed that loss of Lpp particles caused drastic alterations in FB pLDs, indicating a functional connection between Lpp and pLDs at the cell surface (*Ugrankar et al., 2019*). We propose that Act5C may directly assist in lipid metabolism at this LD:Lpp subdomain, perhaps by providing a scaffold where lipid organization or lipid loading into nascent Lpp particles can occur, but this requires future study. It is notable that loss of the cortical actin meshwork in Act5C-depleted cells in the FLP/FRT mosaic FB led to drastic perturbation of the pLD layer, indicating that this actin meshwork organizes and maintains small pLDs near the cell surface, potentially positioning pLDs for interactions with Lpp.

It should also be noted that reduced insulin signaling and feeding in Act5C-deficient larvae may also hamper normal growth and fat storage. Lipodystrophy and insulin resistance often go hand-in-hand

(as widely observed in mammals; *Péterfy et al., 2001*), and Lpp trafficking and insulin signaling are closely coupled in *Drosophila* (*Brankatschk et al., 2014*). An emerging model is that Act5C is required not only for FB-mediated Lpp nutrient signaling, but also for Lpp-mediated lipid transport to the brain to promote insulin secretion and signaling, and subsequent adipose tissue development and larval growth. Thus, our data support a model where blunted lipoprotein trafficking and insulin signaling leads to decreased lipogenesis in the FB. Indeed, lower insulin signaling is associated with reduced lipogenesis in mammals (*Wong and Sul, 2010*). Future studies will dissect how these pathways inter-relate. It should also be noted that genetic suppression of the insulin Receptor (InR) leads to similar developmental arrest as Act5C depletion in the FB (*Oldham et al., 2002*). Previous studies indicate that loss of inter-organ lipid transport (ApoLpp) or nutrient signaling (CK1alpha) also lead to smaller sized larvae and developmental arrest in the larval stages (*Palm et al., 2012*; *Ugrankar et al., 2015*), suggesting shared mechanisms between nutrient sensing/storage and FB cell expansion for biomass accruement. Indeed, ApoLpp-deficient larvae also exhibited smaller FB cells that mirrored Act5C loss, suggesting that the decreased FB cell size in Act5C-depleted larvae may also be also through reduced lipid delivery or its associated nutrient/insulin signaling pathways. Further studies will dissect this relationship between actin dynamics and Lpp-dependent nutrient signaling in FB tissue growth.

## Materials and methods
### Materials availability Statement
All materials related to this study are accessible either through this source or via the BioRxiv preprint. Data also available via Dryad.

Please contact Mike.Henne@utsouthwestern.edu regarding any questions.

### Fly cultures and strains
*Drosophila* L3 late feeding larvae were used for larval assays. All *Drosophila* animal research was performed in accordance with NIH recommended policies. Flies were reared and maintained on standard fly food containing cornmeal, yeast, molasses (~5% sugars), and agar. The *Dcg-Gal4* fat body (FB)-specific driver, previously described (*Suh et al., 2008*), and *Dcg-Gal4;tub-Gal80ts* strains were provided by Jonathan M. Graff (UTSW). The *Cg-Gal4;UAS-Dcr2* FB driver was received from Steve Jean (Université de Sherbrooke), and *UAS-Glycogenin-YFP* from Michael Welte (University of Rochester), respectively. *UAS-Act5C^{RNAi}* and other TRiP RNAi stocks, *Gal4* drivers, *hs-FLP*, and protein/organelle marker strains used were obtained from the Bloomington Stock Center (Bloomington, IN). We obtained additional *Act5C^{RNAi}* strains from Vienna *Drosophila* Research Center, and FRT line for mosaic analysis from Kyoto *Drosophila* Stock Center. The *UAS-hGlut4-GFP* line was made by Best-Gene by injecting a UAS plasmid that was a kind gift of Dr. Leslie Pick (University of Maryland). All Gal4/UAS crosses were cultured at 25 °C to maintain uniform transgene expression. Crosses involving *Cg-Gal4;UAS-Dcr2* with RNAi lines were cultured at 29 °C to maximize knockdown efficiency. A list of genetic tools is available in *Supplementary file 1*.

To track development of *Act5C^{RNAi}* and other FB-specific RNAi strains (*Supplementary file 2*), we crossed 20–25 FB-driver (*Dcg>* or *Cg>*) females with 7–8 *UAS-RNAi* males for ≤24 hr (to ensure similar egg densities across the genotypes examined). Life cycles of these RNAi progeny were closely monitored, and the developmental timeline/ status was recorded.

### Fat body/larval gut dissection and staining
Late feeding L3 larvae were gently removed from inside the food media using a paintbrush and rinsed in water to remove food particles. Larval fat bodies (FBs) or guts were dissected in PBS using Dumont #5 forceps (Electron Microscopy Sciences). All tissues were fixed in 4% paraformaldehyde for 20 min at room temperature and rinsed briefly in PBS prior to staining with organelle labeling dyes. Lipid droplets in FBs were stained by incubating FBs in 100 µM monodansylpentane (MDH) LD stain (Abgent) for 30 min at room temperature, in the dark. F-actin structures were stained with Phalloidin (1:400) or Cell Mask Green Actin Tracking Stain (1:1000) according to manufacturer's instructions (Invitrogen). LDs in guts were stained with 1 µM Nile Red for 30 min at room temperature, in the dark. Next, FBs and guts were rinsed in PBS and mounted on slides in SlowFade Gold antifade reagent with DAPI (Invitrogen) for subsequent imaging.

## Temperature-shift Experiments using tub-Gal80ts

Mated flies (*Dcg-Gal4;tub-Gal80ts X UAS-Act5C^RNAi*) were allowed to lay eggs for 8 hr at 18 °C (permissive temperature) or 4 hr at 29 °C (nonpermissive temperature) and allowed to develop for 1/2/3/4/5 days before moving culture vial from 18° to 29 °C or vice versa for the remainder of fly development. Developmental progress was closely monitored on a daily basis, and larvae from both temperature-shift set-ups were harvested on different days for FB dissection/confocal microscopy (MDH, phalloidin), and TG measurements by TLCs.

## *Drosophila* tissue immunofluorescence

FBs dissected from larvae were fixed in 4% paraformaldehyde at room temperature for 30 min on a nutator, then washed four times with 1 X PBST (PBS in 0.1% Tween), 10 min each. Tissues were blocked in 1 X PBST +2% normal goat serum for 2 hr at room temperature, then incubated with Rb-ApoII primary antibody (1:500 dilution, provided by Dr. Akhila Rajan) added to the blocking buffer, overnight at 4 °C. Next, FBs were washed 4 times with 1 X PBST 15 min each, followed by incubation with anti-rabbit Alexa546 secondary antibody (1:1000 dilution) in blocking buffer at room temperature in the dark for 2 hr on a nutator. FBs were rinsed again four times in 1 X PBST 15 min each, and subsequently mounted in SlowFade Gold antifade reagent with DAPI. Fluorescent structures were observed and imaged by confocal microscopy using standard RFP channel filter at 1 μm step-sizes.

## Confocal fluorescence microscopy

Prepared slides were imaged using Zeiss LSM880 inverted laser scanning confocal microscope. Tissues stained with organelle dyes and those expressing transgenic fluorescent-tagged proteins (e.g. *UAS-mcd8-GFP, UAS-Act5C-RFP, UAS-hGlut4-GFP*) were imaged using the appropriate channel filter DAPI/RFP/GFP. Most FBs were imaged using a ×40 oil immersion objective, as 1 μm z-stack sections, or at lower magnification using a ×10 objective. Larval guts were imaged using a ×20 objective.

## Epifluorescence and other light microscopy

Whole larvae whose FBs were illuminated by FB-expressing fluorescent-GFP markers were imaged using EVOS FL Cell Imaging System (ThermoFisher) at low ×2.5 magnification. Pupal imaging was performed on Leica S8AP0 stereoscope.

## Thin layer chromatography (TLC)

Larvae were weighed, then homogenized in 2:1:0.8 of methanol:chloroform:water. Samples were incubated in a 37 °C water bath for 1 hr. Chloroform and 1 M KCl (1:1) were added to the sample, centrifuged at 3000 rpm for 2 min, and the bottom layer containing lipids was aspirated using a syringe. Lipids were dried using argon gas and resuspended in chloroform (100 μl of chloroform/7 mg of fly weight). Extracted lipids alongside serially diluted standard neutral lipids of known concentrations were separated on TLC plates using hexane:diethyl ether:acetic acid solvent (80:20:1, v/v/v). TLC plate was air dried for 10 min, spray stained with 3% copper (II) acetate in 8% phosphoric acid and incubated at 145 °C in the oven for 30 min to 1 hr to allow bands to develop for scanning and imaging. Neutral lipid (TAG and DAG) band intensity was quantified using Fiji/ImageJ software, and lipid concentrations were calculated from the standard curve generated with standard mixture. In the case of dissected larval guts, dried lipids for all samples were re-suspended in 120 μl of chloroform and final lipid concentrations were calculated by normalizing to protein in sample, measured by standard Bradford assay. Hemolymph (4 μl/sample) for DAG measurements collected from larvae was diluted in 0.1% N-Phenylthiourea (Sigma-Aldrich) in 50 μl PBS. The same lipid extraction protocol was followed, dried lipids were re-suspended in 75 μl of chloroform, and equal volumes were loaded on TLC plates.

## Act5C mutant fat body mosaics

We generated homozygous Act5C mutant clones in the fat body using the FLP/FRT-mediated mitotic recombination system (**Theodosiou and Xu, 1998**). Briefly, *P{lacW}Act5CG0025 P{neoFRT}19 A/FM7c* females carrying the Act5C mutation and Flippase Recognition Target (FRT) sites were crossed with *P{Ubi-mRFP.nls}1, w*, P{hsFLP}12 P{neoFRT}19 A* males that contain heat-shock inducible flippase (FLP) recombinase, a similarly positioned FRT site, and an RFP cell marker. After egg laying, 0–8 hr embryos were heat shocked at 37 °C for 1 hr to induce mitotic recombination between the FRT sites

on homologous chromosome arms. Following heat shock, embryos were allowed to develop normally at 25 °C. Feeding third instar larvae were harvested for fat body dissection and stained with monodansylpentane (MDH) LD stain and Cell Mask Green Actin Tracking stain, or phalloidin (no MDH). Since the cell marker segregates with the wild-type gene, KO cells with 2 copies of mutant Act5C are recognized by the lack of nuclear RFP. Mosaic fat bodies were imaged by confocal microscopy using DAPI (MDH-LDs), GFP (Cell Mask Actin), and RFP (phalloidin and cell marker) filters.

### Quantitative PCR

Whole RNA was extracted from FB tissues using Trizol reagent (Ambion Life Technologies) according to manufacturer's instructions. RNA was quantified by Denovix spectrophotometer, and 1 mg of RNA was reverse transcribed to cDNA using ReadyScript cDNA Synthesis Mix (Sigma). QPCR was performed with cDNA and SYBR Green JumpStart TAQ ReadyMix on the BioRad CFX96 Real-Time System; mRNA expression data were normalized to that of the fly housekeeping gene *rp49*. Primer sequences for all transcripts amplified are available in the supplement data file for primers (table ***Supplementary file 3***).

### Transmission electron microscopy

FB tissue was removed from *Drosophila* larvae, fixed in 2.5% (v/v) glutaraldehyde in 0.1 M sodium cacodylate buffer paraformaldehyde, and processed in the UT Southwestern Electron Microscopy Core Facility. They were post-fixed in 1% osmium tetroxide and 0.8% K3[Fe(CN6)] in 0.1 M sodium cacodylate buffer for 1 hour at room temperature. Cells were rinsed with water and en bloc stained with 2% aqueous uranyl acetate overnight. Next, they were rinsed in buffer and dehydrated with increasing concentration of ethanol, infiltrated with Embed-812 resin and polymerized in a 60 °C oven overnight. Blocks were sectioned with a diamond knife (Diatome) on a Leica Ultracut UCT (7) ultramicrotome (Leica Microsystems) and collected onto copper grids, post stained with 2% aqueous uranyl acetate and lead citrate. Images were acquired on a JOEL 1400 Plus transmission electron microscope using a voltage of 120 kV.

### Feeding assays

Duplicate samples of 3 L feeding larvae (~100 each) were transferred to 9-well plates containing 20% sucrose +Coomassie blue dye (200 µg/ml). After 1 hour of feeding on the dyed sucrose, larvae from the first well were removed and rinsed in water. Larval guts were visualized using a dissecting scope and scored for the presence or absence of Coomassie dye. Larvae in the second well were allowed to feed for an additional 3 hr (total 4 hr), before removing from sucrose, washing, and scoring guts for food consumption. To quantify relative feeding,~size-matched larvae with visually detectable gut dye were washed in water, then rinsed briefly in methanol, blotted dry, and transferred to 1.5 ml chilled Eppendorf tubes (30 larvae/tube, in triplicate). Larvae were homogenized in 200 µl of 100% methanol. The homogenate was centrifuged at maximum speed for 10 min at 4 °C. Supernatant was transferred to fresh tubes containing equal volume of dH2O, and centrifuged again. Absorbance of supernatant was measured at 595 nm, and dye ingested was determined by comparing to a standard curve of Coomassie dye concentrations.

### 2-NBD-glucose (2-NBDG) and BODIPY-C12 uptake assays

FB tissues dissected from L3 feeding larvae starved in PBS for 4 hr (or FB tissues extracted from larvae and pre-incubated in PBS for 4 hr) to deplete glycogen stores, were incubated in 300 µM of a fluorescently labeled glucose analog 2-NBDG (2-Deoxy-2-[(7-nitro-2,1,3-benzoxadiazol-4-yl)amino]-D-glucose) (Sigma-Aldrich, #72987) for 45 min at room temperature, in the dark. FBs were then washed with PBS, fixed in 4% paraformaldehyde for 20 min, washed again in PBS, and mounted in SlowFade on slides. 2-NBDG was imaged using the GFP channel filter by confocal microscopy. To measure BODIPY-C12 uptake, FB tissues from larvae were incubated in BODIPY FL C12 (4,4-Difluoro-5,7-Dimethyl-4-Bora-3a,4a-Diaza-s-Indacene-3-Dodecanoic Acid) (Invitrogen, #D3822) 1:1000 in PBS for 20 min at room temperature, in the dark. Subsequent to fixation with 4% paraformaldehyde, FBs were also stained with MDH LD dye to visualize if BODIPY-C12 was incorporated into LDs. Prepared slides were imaged using DAPI(MDH), and RFP(BODIPY-C12) channel filters on the confocal microscope. Both 2-NBDG and BODIPY-C12 FB cell fluorescence was quantified using the Fiji/Image J software.

## Hemolymph glucose and trehalose assays

Feeding larvae washed out from culture vials were collected in 630 µm mesh-fitted baskets (Genesee) and rinsed to get rid of adherent food particles. Larvae were dried and divided onto into piles (10–12 larvae each) on a strip of parafilm. Larvae were bled by tearing the cuticle with Dumont 5 forceps (Electron Microscopy Sciences). Two µl of colorless hemolymph was aspirated from each pile and separately transferred to 96-well plates (Thermo-Scientific) containing 0.1% N-Phenylthiourea (Sigma-Aldrich) in 50 µl PBS. 150 µl of Autokit Glucose reagent (Wako) was added to each well and incubated at room temperature for 20 min before measuring absorbance at 505 nm. Glucose concentration was calculated from a standard curve generated with manufacturer's glucose standards. For trehalose assays, 8 µl of dilute hemolymph was treated with 5 µl of (diluted 8 X) porcine kidney trehalase (Sigma) overnight at 37 °C. Ten µl of treated sample was assayed for trehalose as described for glucose.

## Glycogen assay

Dissected FBs were homogenized in 300 µl ice-cold PBS using sonication followed by a syringe (29G1/2 needle). After reserving 20 µl of the homogenate for protein Bradford quantification, the rest of the homogenate was heat inactivated at 70 °C for 10 min. Homogenate was then centrifuged at maximum speed at 4 °C for 10 min and supernatant was collected in a new tube. In a 96-well plate, 30 µl of each sample was loaded in duplicate rows. Then, 100 µl of Autokit Glucose reagent +amyloglucosidase (1 µl amyloglucosidase {Sigma A1602; 25mg} per 1 ml of Glucose reagent) was added to one row of samples, and 100 µl of Glucose reagent alone (without amyloglucosidase) was added to the duplicate row of samples, and to the glucose standards. The plate was incubated at 37 °C for 30 min, after which color intensity was measured using a microplate reader at 505 nm. Free glucose concentration in treated and untreated samples was calculated based on the glucose standard curve. Glycogen concentration (as free glucose) was determined by subtracting glucose in the untreated samples from those treated with amyloglucosidase. Finally, total glycogen was normalized to protein measured by the Bradford assay.

## Fat body glycogen staining

Glycogen in FB tissues was stained using the Periodic Acid-Schiff (PAS) kit (Sigma). Larval FBs were dissected in 1% BSA in PBS, fixed with 4% paraformaldehyde for 20 min, and washed twice with 1% BSA in PBS. FBs were then incubated in 0.5% period acid solution for 5 min, washed twice with 1% BSA in PBS, then stained with Schiff's reagent for 15 min, washed again, and mounted on slides. PAS-stained FB tissues were imaged using the Leica DM6B light microscope.

## Western blotting

Feeding larvae were bled on parafilm, and 20 µl of pooled hemolymph was collected per sample and transferred to a chilled 1.5 ml Eppendorf tube containing 40 µl of Ringer's+protease inhibitor cocktail. Samples were centrifuged at 300 X $g$ for 10 min at 4 °C to pellet hemocytes. The supernatant was transferred to fresh tubes on ice. Sample for loading was prepared by mixing 20 µl of dilute hemolymph (equal volumes for each sample) with 10 µl of 4 X LDS buffer containing 0.2 M DTT and 10 µl Ringer's solution, and boiling at 95 °C for 5 min. Sample was run on a 10% pre-cast polyacrylamide gel (BioRad) at 120 V for 1 hr. Gel proteins were transferred to nitrocellulose membrane by wet transfer. After briefly rinsing membrane in Milli-Q water, Ponceau stain was added to blot for 15 min, then rinsed again with MQ-water, and proteins on membrane imaged using BioRad Imager. Membrane was blocked with 5% Casein in 1 X TBS-T (0.1% Tween) for 1 hr, then incubated overnight at 4 °C with rabbit primary antibody against ApoII (provided by Dr. Akhila Rajan) at a 1:1000 dilution. After washing with TBS-T (4 X for 5 min), membrane was incubated with HRP-conjugated goat anti-rabbit secondary antibody (Abcam ab6721) (1:5000 dilution) for 1 hr at room temperature. Membrane was washed in TBS-T (4 X for 5 min), developed with Clarity Western ECL blotting substrate (Bio-Rad, ab1705061) and imaged with BioRad Imager.

For the DILP2 immunoblot, a 4 to 20% polyacrylamide gel was employed due to the low molecular weight of DILP2 (12 kDa). Further, LI-COR secondary antibodies (1:5000 dilution) were used against chicken DILP2 (1:1000 dilution, provided by Dr. Akhila Rajan). Here we used Immobilon PVDF-FL PVDF membrane (Millipore Sigma, IPFL0010) activated in methanol for 5 min, and 1% fish gelatin in

1 X TBS as blocking buffer. Secondary antibody used was donkey anti-chicken IRDye 800CW (LI-COR, 926–32218).

## Mammalian cell culture

Huh7 cells were cultured in DMEM (Corning; 10–017-CV) supplemented with 10% Fetal calf serum (Corning; 35–010-CV) and 1% penicillin streptomycin solution (Corning; 30–002- Cl). To induce lipid droplet biogenesis, cells were incubated with 600 µM of Oleic acid (Sigma Aldrich; O1008) conjugated with 100 µM of fatty acid-free BSA (Sigma-Aldrich; A3803) for 16 hr.

## Mammalian cell immunofluorescence (IF) staining and confocal microscopy

After OA treatment, the cells were fixed with 4% paraformaldehyde solution in PBS for 15 min at room temperature. For IF staining, fixed cells were washed with 1 x PBS and permeabilized with 0.1% Triton X-100 in 1 X PBS at room temperature for 5 min. The permeabilized cells were blocked in IF buffer (PBS containing 3% BSA and 0.1% Triton X-100) for 30 min. The cells were then incubated with primary antibody in IF buffer for 1 hr and washed 3 X with PBS. This was followed by incubation with secondary antibody in IF buffer for 30 min and 3 X washes with PBS. The primary antibodies used are goat anti-Apolipoprotein B (1:200; Rockland Immunochemicals, Inc; 600-101-111), and mouse anti-Hsp90B1 (1:100; Sigma Aldrich; AMAb91019). The secondary antibodies are donkey anti-mouse AF594 (ThermoFisher Scientific; R37115), and donkey anti-goat AF488 (ThermoFisher Scientific; A11055) used at a dilution of 1:400. Visualization of LDs were achieved by staining the cells with MDH (1:1,000; Abgent; SM1000a) for 15 min at room temperature. The images of the cells were taken using a ×63 oil-immersion objective in a laser scanning confocal Zeiss LSM880+Airyscan microscope. Fiji/ImageJ was used for the representation and quantification of images.

## Image analysis and quantification

Fiji/ImageJ was used for quantifications involving size of FB lipid droplets and area of larval fat cells imaged by confocal microscopy. First, the area was measured for the entire field of view or a region-of-interest (ROI) of defined size. For lipid droplets (LDs) size measurements, prior to converting image to binary, a 'smooth' function was applied to the image to remove inherent graininess of the lipid stain and allow for more accurate quantification of LD cross-sectional area. The method for binary conversion was selected based on which one best represented LDs in the region of interest. 'watershed' was performed on binary images, and 'analyze particles' was used to quantify LD number and size. For fat cell area, the 'polygon selections' tool was used to draw boundaries around individual cells highlighted by fluorescent protein-tagged markers (e.g. phalloidin, Cell Mask, mcd8-GFP, hGlut4-GFP), and 'analyze', 'measure area' functions were applied. All volumetric analysis and multi-color linescans were conducted using Fiji/ImageJ software.

## Statistical analysis

All experiments were conducted at least three times to provide experimental rigor. For the figures, all comparative statistics were either student t-tests or ANOVA tests for multiple data point experiments. Statistical significance was applied if $p < 0.05$. Error bars indicate SD. In general a single * indicates $p < 0.05$, ** indicates $p < 0.01$, *** indicates $p < 0.001$, **** indicates $p < 0.0001$.

## Acknowledgements

The authors thank members of the Henne and Friedman labs for insights and discussion. We also thank Dr. Akhila Rajan for the kind use of her anti-ApoLII Lpp and anti-DILP2 antibodies, Dr. Michael Welte for the UAS-Glycogenin-YFP line, and Dr. Steve Jean (Université de Sherbrooke) for the Cg-Gal4;UAS-Dcr2 driver. We would also like to thank Dr. Anza Darehshouri, Phoebe Doss, and Rebecca Jackson for assistance with TEM. We are also grateful to Dr. Saikat Mukhopadhyay for use of his light and stereo microscopes and Real-Time QPCR machine. WMH is supported by funds from the Welch Foundation (I-1873), the NIH NIGMS (GM119768), NIDDK (DK126887), and the UT Southwestern Endowed Scholars Program.

## Additional information

### Funding

| Funder | Grant reference number | Author |
|---|---|---|
| National Institute of Diabetes and Digestive and Kidney Diseases | DK126887 | Rupali Ugrankar-Banerjee<br>Son Tran<br>Anastasiia Kovalenko<br>Blessy Paul<br>W Mike Henne |
| National Institute of General Medical Sciences | GM119768 | W Mike Henne |
| Welch Foundation | I-1873 | W Mike Henne |
| University of Texas Southwestern Medical Center | Endowed Scholars Program | W Mike Henne |

The funders had no role in study design, data collection and interpretation, or the decision to submit the work for publication.

### Author contributions

Rupali Ugrankar-Banerjee, Conceptualization, Data curation, Formal analysis, Investigation, Visualization, Methodology, Writing - original draft, Project administration; Son Tran, Conceptualization, Data curation, Methodology; Jade Bowerman, Data curation; Anastasiia Kovalenko, Data curation, Formal analysis, Methodology; Blessy Paul, Data curation, Methodology; W Mike Henne, Conceptualization, Data curation, Formal analysis, Supervision, Funding acquisition, Validation, Investigation, Writing - original draft, Project administration

### Author ORCIDs

W Mike Henne  http://orcid.org/0000-0002-2135-2799

### Decision letter and Author response

Decision letter https://doi.org/10.7554/eLife.81170.sa1
Author response https://doi.org/10.7554/eLife.81170.sa2

## Additional files

### Supplementary files

• Supplementary file 1. *Drosophila* lines and tools. Table of *Drosophila* lines used in this study.

• Supplementary file 2. Table of developmental phenotypes. Table of developmental phenotypes associated with different *Drosophila* lines used in this study.

• Supplementary file 3. Oligos table. Table of oligos used in this study.

• MDAR checklist

### Data availability

We have now provided raw datasets for the manuscript on a Dryad file site (https://doi.org/10.5061/dryad.4mw6m90d4).

The following dataset was generated:

| Author(s) | Year | Dataset title | Dataset URL | Database and Identifier |
|---|---|---|---|---|
| Ugrankar R, Tran S, Bowerman J, Kovalenko A, Paul B, Henne M | 2023 | Data from: The fat body cortical actin network regulates *Drosophila* inter-organ nutrient trafficking, signaling, and adipocyte cell size | https://dx.doi.org/10.5061/dryad.4mw6m90d4 | Dryad Digital Repository, 10.5061/dryad.4mw6m90d4 |

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
