## [Editor Report]

This important study defines the involvement of the actin cytoskeleton in adipocyte function. The authors present convincing evidence that Actin 5C is a critical mediator of lipid metabolism, nutrient uptake, and larval development in *Drosophila*. This work provides novel insight into lipid metabolism, having broad implications in multiple fields.

---

## [Decision Letter]

**Decision letter after peer review:**

Thank you for submitting your article "The fat body cortical actin network regulates *Drosophila* inter-organ nutrient trafficking and adipocyte cell size" for consideration by *eLife*. Your article has been reviewed by 2 peer reviewers, and the evaluation has been overseen by a Reviewing Editor and Utpal Banerjee as the Senior Editor. The reviewers have opted to remain anonymous.

Essential revisions:

1) Both reviewers have concerns about lack of the RNAi KO efficiency, the lack of testing independent RNAi and the lack of using available mutants. These data are important as the conclusions need to be based on compelling genetic data.

2) Please demonstrate that the food intake for the control vs actin 5C RNAi larvae is similar before any conclusions can be made about nutrient uptake.

3) Consider the potential maternal contribution of actin 5C to the embryos before you conclude that it is not required during embryogenesis or early larval development. Alternatively, do not make this statement.

4) Address the issues raised about the DCG-GAL4 driver

5) Determine if insulin signalling is affected?

*Reviewer #1 (Recommendations for the authors):*

1. The knockdown efficiency for all RNAi need to be validated by qPCR or, ideally, by WB when there is an available antibody. Further, key data need to be repeated using a second, independent RNAi per gene to show the effects observed are specific to the gene target and not an off-target effect of the RNAi. If no additional RNAi exist, mutant (clonal to avoid lethality?) fly lines could also be used to show consistent data can be obtained between RNAi and LOF mutations.

2. The data ruling out the role of additional actin isoforms could be the result of different knockdown efficiencies with the different RNAi vs a biological impact. Thus, it is critical to understand the knockdown efficiency of these RNAi. Further, there could be compensation here. Thus, all isoforms should be assayed in animals expressing a single RNAi to make sure that the lack of an effect is not due to upregulation of a not-targeted, compensatory isoform.

3. Spectrin can interact with actin. Thus, can the authors demonstrate that spectrin RNAi is acting independently of the actin network as suggested in the text? If possible, co-staining for phallodin and spectrin with an antibody or using a spectrin reporter line (e.g. UAS-spectrin-GFP) would also be informative. Overall, these data would help the authors/readers to understand the difference in the periphery vs midline LD impacts for the actin vs spectrin RNAi. In particular, the authors state that spectrin is localized to the cell surface so it could simply be that spectrin loss disrupts actin at the cell surface but not the midline due to its localization versus, as argued in the text, spectrin acting independent of actin 5C.

4. In Figure 4, the authors use nutrient uptake assays to ask if actin 5C loss in FB altered carbohydrate and FA uptake. The authors need to demonstrate that the food intake for the control vs actin 5C RNAi larvae is similar before any conclusions can be made about nutrient uptake. Of particular concern is the data in Figure 2 showing a dramatic difference in size, development speed, and viability in control vs actin 5C RNAi larvae. Thus food intake and capacity is likely to differ dramatically between these genotypes. This will also impact conclusions made from data in Figure 5.

5. The use of the GAL80[TS] system and temperature shifts to address the need for actin 5C at different developmental stages is nice (Figure 6). However, the authors need to consider the potential maternal contribution of actin 5C to the embryos before they can conclude that it is not required during embryogenesis or early larval development as maternally derived actin 5C may be compensating here.

*Reviewer #2 (Recommendations for the authors):*

Specific questions and additional experimental suggestions:

1. The authors use RNAi-based knockdown to study the role of Act5c in fat tissue. To increase the rigor of the work, the authors would need to utilize other orthogonal genetic tools. As I believe further lines of evidence are required to substantiate whether Act5c is indeed performing specific functions in lipoprotein localization, Glut localization and required to show that these genetic tools (Act5c-RNAi and Dcg-Gal4) are indeed robust. I outline some specific experiments and request clarifications below.

a. Have the authors used multiple independent RNAi lines for Act5c-RNAi? If yes, then it would be best if they specify this in the methods with a proper table.

b. What is the KD level – by qPCR – of Act5c and the other actin isoforms? To exclude the role of the other isoforms, they should use independent RNAi lines in figure 2B, and show (using qPCR) that these other isoforms are knocked down.

c. The authors use Dcg-Gal4. While it is clear from this study and their 2019 study that Dcg-Gal4 does express in fat, there is no clear characterization of whether Dcg-Gal4 is expressed in other tissues such as the gut, muscle, or other tissues. In their 2019 study, the authors cite the Suh JM 2008 study as the source of Dcg-Gal4. On examining the 22008 papers, I could not find any evidence on the characterization of Dcg-Gal4. The question remains, what is the specificity of its expression in the fat tissue? Does it not express in n the gut, brain, or muscle? Especially to conclude the gut-fat signaling (Figure 5), the authors would need to exclude the expression of Dcg-Gal4 in the gut or use other fat-body drivers which do not express in the gut, such as Cg-Gal4 or Lpp-Gal4.

d. m concerned about drawing strong conclusions from RNAi -KD studies without using Genetic alllesm especially when they are readily available. Even if the authors use multiple independent RNAi's for Act5c, it would be important to use Act5c mutant alleles to corroborate their findings. I appreciate that the Act5c alleles are embryonic lethal. But these alleles are available on the Kyoto stock center as recombined on an FRT (FRT19A) [111764-Act5CG0025 P{ry+t7.2=neoFRT}19A / FM7c; P{ey-FLP.N}5]. Hence, the authors should consider making clones using hsFLP or leaky fat body flippase drivers (see Figure 5; PMID: 19562034 or Figure 4; PMID: 29694367) to make clones of Act5c and then assess- what happens to pLD, mLD; hGLut4 trafficking (Figure 4K) etc.,. This is especially critical for datasets where they over-express hGLut4 (4K) but then KD using RNAi Act5c and conclude that Act5C does not affect trafficking of hGlut4. It is possible that by overexpressing hGlut4 GFP in a hypomorphic condition (RNAi KD), the effect of Act5c in such events is masked. Thus, by using clonal analysis of existing FRT recombined Act5C alleles (from Wagner, 2002- available in stock centers Kyoto), the authors could bolster their conclusions.

2. The data on fat-gut lipophorin trafficking is intriguing. Nonetheless, the authors would need to ensure that the Dcg-Gal4 is not expressed in the gut and is fat specific (see point 1c). Furthermore, they need to clarify whether indeed Apolpp secretion is affected. The images of just the Apolpp staining Figure 5E and 5F (TEM) are insufficient evidence to draw a strong conclusion. They can do this by performing western blot with the Apolpp antibodies on hemolymph from the fat-specific Act5CRNAi, as has been shown in Palm 2012 (PMID: 22844248).

3. Overall, the phenotypes that the author show for Dcg-Gal4>UAS Act5c RNAi- reduced cell size, reduced lipolysis and growth defects. These are all consistent with a defect in insulin signaling. Previous studies have shown that defective Apolpp signaling leads to defective insulin signaling (PMID: 25275323). To increase impact and strengthen the connection between Act5c Apolpp, it would benefit the study, if the authors test whether there is indeed reduced insulin activity in the Act5CRNAi. They can do this by testing p-Akt levels, and tGPH reporter activity.

4. The authors have investigated Apolpp, but ApoLTP is another transporter specifically between the fat and gut (Palm et al., 2012; PMID: 22844248). Hence, the authors are encouraged to examine ApoLTP localization, and perform experiments with Apoltp.

Presentation:

1. The authors show that Tango phenocopies Act5C. That is in line with their observation that Apolpp is affected. Therefore it is puzzling that the authors do not cite or discuss the study in 2016 on the role of Tango in ApoB secretion (PMID: 27138255). Similarly, the authors show GRASP puncta and say they affect Golgi secretion. But GRASP could be affecting non-conventional secretion of lipophorins, since it has been shown to affect non-conventional secretion in *Drosophila* (PMID: 18267086 and 29017032). Similarly, the authors should discuss the known role of lipophorin signaling on insulin signaling (PMID: 25275323). From a presentation point of view, the manuscript would benefit from contextualizing observations on secretion and lipophorins with prior studies.

2. The current presentation, makes the final conclusions diffuse. Specifically, the last two figures characterize Act5c KD effects on larval development. The manuscript would be more focused if Figure 6 can be moved to supplement and Figure 7 be moved after Figure 2, and the portions with Apolpp RNAi can be presented in an Apolpp specific figure.

---

## [Author Response]

Essential revisions:1) Both reviewers have concerns about lack of the RNAi KO efficiency, the lack of testing independent RNAi and the lack of using available mutants. These data are important as the conclusions need to be based on compelling genetic data.

We thank the reviewers for these valuable comments. We have done several experiments to further validate the RNAi/KD of Act5C:

1. To rule out off-target effects for the RNAi phenotypes, we have now tested two other independent RNAi lines for Act5C obtained from VDRC (VDRC Act5C lines #7139 and #101438). Both show the same results as our original Act5C-RNAi line obtained from Bloomington—that FB-tissue specific KD by these RNAi lines leads to fat loss and perturbed LD morphology (Figure Supplement 2), as well as small FB cells and reduced phalloidin staining at the FB cell surface (Figure Supplement 1). We also measured TG levels in the Act5C VDRC#7139 tissue and found it was also decreased by very similar levels from the original Act5C RNAi line (Figure Supplement 2). Thus, all three Act5C RNAi lines independently show that Act5C depletion leads to reduced LD and fat storage in the FB tissue, as well as reduced cell size.

2. Per the reviewer requests, we also utilized FLP/FRT methodology to RNAi deplete Act5C in individual FB cells (now in Figure 2). Strikingly, this revealed that individual FB cells within a tissue mosaic that lacked Act5C displayed perturbed LD morphology and reduces LD density. pLDs were also largely missing in these Act5C depleted cells, and appeared to have collapsed/fused into a fat mass at the cell periphery (Figure 2). This supports a model where Act5C is necessary to organize and maintain the pLD layer at the cell periphery.

3. To confirm the Act5C RNAi KD was indeed targeting Act5C, we have now conducted qPCR analysis. We find that Act5C RNAi in the FB leads to a ~70% depletion of Act5C expression (Figure Supplement 1). In line with this, phalloidin staining in Act5C RNAi FB tissue confirms significant F-actin loss (Figure Supplement 1). This, combined with the independent RNAi lines mentioned in point 1 above indicate that Act5C RNAi-depletion leads to decreased fat storage and smaller cell size.

4. To confirm that the Act5C RNAi was specifically targeting Act5C and not other actins, we developed an imaging-based system to monitor GFP-tagged actin expression during UAS-RNAi treatment. We find that Act5C RNAi depleted Act5C-GFP in the FB, but not any other Actin isoforms (i.e. other ActXX-GFP UAS lines) (Figure Supplement 1). In line with this, Act5C RNAi was the only line to deplete the cortical actin phalloidin stain in the FB. This strongly suggests that Act5C is the primary actin isoform composing the FB cortical actin network.

2) Please demonstrate that the food intake for the control vs actin 5C RNAi larvae is similar before any conclusions can be made about nutrient uptake.

Thank you for this suggestion. We actually find that, while Act5C deficient larvae can feed, they generally feed slower than control larvae (quantified in Figure 6). This is actually an unexpected and exciting result in our study, and we further investigated this phenomenon, as seen in the new Figure 6. We find that FB-specific Act5C-depleted larvae have smaller FB tissues, which correlates with reduced insulin signaling. We monitored insulin signaling in several ways: (1) via qPCR analysis, (2) via a PIP3 fluorescent biosensor tGPH, and (3) by direct Western blotting of hemolymph DILP2 (Figure 6). All these methods independently indicate blunted insulin signaling in Act5C-depleted larvae. While a full dissection of this is beyond the scope of the present study, we hypothesize that loss of the Act5C cortical actin network causes defects in adipose tissue cell size expansion during development, as well as defective lipoprotein trafficking, which correlates with blunted insulin signaling. Indeed, numerous past and even recent studies indicate that insulin and lipoprotein signaling are tightly coupled (Kelly, e*Life*, 2022), and our collective data are also consistent with this model.

It is also important to note that Act5C-deficient FBs have normal glycogen carbohydrate storage (Figure 7). As most triglycerides are made from de novo lipogenesis in the FB, this suggests that nutrients are not limiting in Act5C-deficient animals for LD biogenesis and fat storage. Rather, we hypothesize that Act5C loss results in altered nutrient signaling and Lpp trafficking. This will be a topic of future investigation.

3) Consider the potential maternal contribution of actin 5C to the embryos before you conclude that it is not required during embryogenesis or early larval development. Alternatively, do not make this statement.

This is an excellent point, and we have now amended the text to address this. We now state that we cannot rule out that maternally inherited Act5C plays a role in the early stages of embryogenesis (Timothy C. Burn et al., 1989).

4) Address the issues raised about the DCG-GAL4 driver

We have addressed this in two ways:

1. We now show that the Dcg-Gal4 driver is specific for the FB, and does not express in other tissues such as the gut and brain (Figure Supplement 1).

2. We now show that other Act5C RNAi lines driven by other FB-specific drivers including Cg-Gal4 and Lpp-Gal4 have the same phenotypes, including reduced lipid/LD storage and decreased cell size (Figure Supplement 2).

5) Determine if insulin signalling is affected?

Thank you, this is an excellent suggestion. We have done several experiments to address insulin signaling, which is linked to a new Figure present in this revised manuscript:

1. We now show that insulin signaling is down in Act5C-depleted FBs via expression of the PI3K fluorescent biosensor tGPH (Figure 6).

2. We also directly monitor larval hemolymph DILP2 insulin levels, showing they are significantly lower in larvae with FB-specific Act5C depletion (Figure 6).

3. In line with lower insulin/DILP2 signaling, we now show that FB-specific Act5C depletion causes reduced feeding behavior (Figure 6).

4. Relatedly, we now show by qPCR that Tobi, a direct transcriptional target of insulin signaling is downregulated, while expression of ImpL2, which binds and represses circulating DILP2, is increased (Figure 6).

Reviewer #1 (Recommendations for the authors):1. The knockdown efficiency for all RNAi need to be validated by qPCR or, ideally, by WB when there is an available antibody. Further, key data need to be repeated using a second, independent RNAi per gene to show the effects observed are specific to the gene target and not an off-target effect of the RNAi. If no additional RNAi exist, mutant (clonal to avoid lethality?) fly lines could also be used to show consistent data can be obtained between RNAi and LOF mutations.

We thank the reviewer for this point. We have done several things to address this issue:

1. First, we generated multiple sets of oligos to validate the KD of actins. We were successfully able to confirm that KD of Act5C RNAi (~70% efficiency) in the FB with our original Act5C RNAi TRiP line we received from Bloomington (Figure Suppl 1). Other qPCR attempts to monitor other actin genes were problematic, potentially due to the high degree of sequence overlap between different actins. To circumvent this, we developed a GFP-based imaging approach to monitor RNAi depletion of actins isoforms (see point 3 below).

2. To rule out off-target effects of our Bloomington Act5C RNAi line, we have now tested two independent Act5C UAS-RNAi lines we received from VDRC. Upon FB-specific KD, both recapitulate the phenotypes observed with the original Act5C RNAi line (reduced LD/lipid storage and smaller FB cell size) (Figure Suppl 1,2). We also monitored TG levels with both these VDRC lines (VDRC#7139 and #101438). We observed significant TG decreases in the FB as with the original Bloomington RNAi line (Figure Suppl 2).

3. To confirm that Act5C RNAi depletion is specific for Act5C and not other actin isoforms, and to overcome the issues faced with qPCR (see point 1 above), we took advantage of GFP-tagged actin lines and microscopy to monitor how Act5C RNAi impacted different actin-GFP expressions. We find that Act5C RNAi specifically reduced Act5C-GFP expression (and altered its distribution) in the FB, but not any other GFP-tagged actins (Figure Suppl 1). This indicates that the Act5C RNAi targets Act5C specifically.

4. We also utilized FLP/FRT mitotic recombination to KO Act5C in subsets of FB cells within a continuous tissue (Figure 2). Consistent with other lines, this shows that loss of Act5C in a cell autonomous manner led to disruption of the cortical cytoskeleton and LD morphology.

2. The data ruling out the role of additional actin isoforms could be the result of different knockdown efficiencies with the different RNAi vs a biological impact. Thus, it is critical to understand the knockdown efficiency of these RNAi. Further, there could be compensation here. Thus, all isoforms should be assayed in animals expressing a single RNAi to make sure that the lack of an effect is not due to upregulation of a not-targeted, compensatory isoform.

Thank you for the point. As stated above, we have now confirmed: (1) that the knockdown in our original Act5C RNAi line in the FB occurs with ~70% efficiency (Figure Suppl 1), (2) that it is specific for Act5C and not other actins, (3) that the defects observed are not due to off-target effects, as two other independent RNAi lines (and a FLP/FRT line) also cause similar cytoskeletal and FB defects (Figure Suppl 1,2; Figure 2) and (4) that effects we see in the FB from loss of Act5C is a FB-driven effect (and not from other tissue KDs) due to using other established FB-specific drivers (Figure Suppl 2).

Additionally we find, using phalloidin F-Actin stain, that Act5C RNAi is the only line that depletes the FB cortical actin network (Figure Suppl 1). We acknowledge that we cannot rule out that the lack of a phenotype from other actin isoform RNAi lines is from inefficient KD of those actins, and as such have amended the text to acknowledge this limitation.

3. Spectrin can interact with actin. Thus, can the authors demonstrate that spectrin RNAi is acting independently of the actin network as suggested in the text? If possible, co-staining for phallodin and spectrin with an antibody or using a spectrin reporter line (e.g. UAS-spectrin-GFP) would also be informative. Overall, these data would help the authors/readers to understand the difference in the periphery vs midline LD impacts for the actin vs spectrin RNAi. In particular, the authors state that spectrin is localized to the cell surface so it could simply be that spectrin loss disrupts actin at the cell surface but not the midline due to its localization versus, as argued in the text, spectrin acting independent of actin 5C.

This is an excellent point. We now show that F-actin can still localize to the cortical actin network in Β-spectrin RNAi-depleted FBs (Figure Suppl 8). We also find that β-spectrin loss does not impact Lpp secretion from the FB, whereas Act5C depletion does (Figure 8). However, we agree that we cannot fully distinguish whether B-spectrin works completely independently from Act5C, and have amended the text.

4. In Figure 4, the authors use nutrient uptake assays to ask if actin 5C loss in FB altered carbohydrate and FA uptake. The authors need to demonstrate that the food intake for the control vs actin 5C RNAi larvae is similar before any conclusions can be made about nutrient uptake. Of particular concern is the data in Figure 2 showing a dramatic difference in size, development speed, and viability in control vs actin 5C RNAi larvae. Thus food intake and capacity is likely to differ dramatically between these genotypes. This will also impact conclusions made from data in Figure 5.

We thank the reviewer for this excellent point. Indeed, we find that FB-specific Act5C depletion leads to reduced feeding as well as altered insulin signaling (Figure 6). This was actually an unexpected and potentially exciting finding. Briefly, we find that Act5C loss correlates with reduced/slowed feeding, as well as reduced DILP2 in the hemolymph, blunted FB insulin signatures from a fluorescent PIP3 biosensor (Figure 6). It is beyond the scope of this study to fully understand these changes, but we postulate that loss of Act5C in the FB leads to smaller adipose tissue, which in turn causes lower lipoprotein secretion and blunted insulin signaling. This will be the subject of future studies.

5. The use of the GAL80[TS] system and temperature shifts to address the need for actin 5C at different developmental stages is nice (Figure 6). However, the authors need to consider the potential maternal contribution of actin 5C to the embryos before they can conclude that it is not required during embryogenesis or early larval development as maternally derived actin 5C may be compensating here.

Thank you for this feedback. We agree that we cannot at present rule out the maternal contribution of Act5C or actins in early embryo development. We have amended the text to address this.

Reviewer #2 (Recommendations for the authors):Specific questions and additional experimental suggestions:1. The authors use RNAi-based knockdown to study the role of Act5c in fat tissue. To increase the rigor of the work, the authors would need to utilize other orthogonal genetic tools. As I believe further lines of evidence are required to substantiate whether Act5c is indeed performing specific functions in lipoprotein localization, Glut localization and required to show that these genetic tools (Act5c-RNAi and Dcg-Gal4) are indeed robust. I outline some specific experiments and request clarifications below.a. Have the authors used multiple independent RNAi lines for Act5c-RNAi? If yes, then it would be best if they specify this in the methods with a proper table.

Thank you for this suggestion. As discussed in depth above for Reviewer 1, we have now used two additional Act5C RNAi lines to deplete Act5C in the fat body, as well as a FLP/FRT approach to generate Act5C mutant cells in FB chimera (Figure Suppl 1 and 2, Figure 2). These new experiments confirm Act5C loss leads to altered LD morphology, decreased TG storage, as well as smaller cell size and reduced phalloidin stain in the FB. We also conduct several qPCR and other assays confirming KD (see point below).

b. What is the KD level – by qPCR – of Act5c and the other actin isoforms? To exclude the role of the other isoforms, they should use independent RNAi lines in figure 2B, and show (using qPCR) that these other isoforms are knocked down.

Thank you, we now report that our original Act5C RNAi TRiP line gives ~70% KD efficiency (Figure Suppl 1). Regarding qPCR of other actin isoforms: Given the high degree of sequence overlap between different actins in *Drosophila*, this was a challenging issue to address. However, we addressed this in a few ways:

1. As mentioned, we have now quantified Act5C KD efficiency for our original Act5C RNAi line, using validated oligo sets (3 independent primer sets) specifically designed for Act5C ( sources are: Ogneva et al., 2016 DOI:10.1371/journal.pone.0166885; Blatt et al., 2021 https://doi.org/10.1016/j.cub.2021.04.052; Borkuti et al., 2022 DOI 10.3389/fmolb.2022.963635). Importantly, we also find that UAS-RNAi knockdown of Act42A (the other major cytoplasmic actin in FB) does not significantly impact Act5C RNAi levels, suggesting specificity of our Act5C knockdown (Figure Suppl 1).

2. We also generated (and obtained published) oligos to monitor all other actin isoforms, but these oligos appeared to cross react with multiple actins and were therefore challenging to interpret. To overcome this issue, we took advantage of the existence of GFP-tagged UAS lines for all *Drosophila* actins, and used these to examine how RNAi depletion of Act5C would impact other Actin-GFP expression. We find that UAS-Act5C-RNAi in the FB impacts the distribution and expression of Act5C-GFP, but of any other Actin-GFP constructs (Figure Suppl 1). This strongly suggests that Act5C RNAi specifically depletes/targets only Act5C in the FB, and supports a model where Act5C is necessary for the FB cortical actin network. However, we cannot rule out that other actins like Act42A may also be necessary for the cortical actin network of the FB, and that we may be only partially depleting these with our RNAi approaches. We have adjusted the text in this revision to reflect this.

c. The authors use Dcg-Gal4. While it is clear from this study and their 2019 study that Dcg-Gal4 does express in fat, there is no clear characterization of whether Dcg-Gal4 is expressed in other tissues such as the gut, muscle, or other tissues. In their 2019 study, the authors cite the Suh JM 2008 study as the source of Dcg-Gal4. On examining the 22008 papers, I could not find any evidence on the characterization of Dcg-Gal4. The question remains, what is the specificity of its expression in the fat tissue? Does it not express in n the gut, brain, or muscle? Especially to conclude the gut-fat signaling (Figure 5), the authors would need to exclude the expression of Dcg-Gal4 in the gut or use other fat-body drivers which do not express in the gut, such as Cg-Gal4 or Lpp-Gal4.

Thank you for this question. We have addressed this concern in two ways:

1. We have now utilized two other FB-specific drivers (Cg-Gal4 and Lpp-Gal4) to determine whether Act5C RNAi depletion gives similar outcomes as when driven by the Dcg-Gal4 driver. Indeed this is the case—we find UAS-RNAi depletion of Act5C in the FB via either Cg-Gal4 or Lpp-Gal4 leads to reduced LD lipid storage and smaller cell size (Figure Suppl 2).

2. We have confirmed that Dcg-Gal4 is FB specific by imaging other tissues including the brain and gut in animals expressing UAS-GFP on the Dcg-Gal4 driver. While we find bright GFP signal in the FB as expected, we find no detectable GFP signal in either the gut nor brain (Figure Suppl 1). This indicates that the Dcg-Gal4 driver is FB-specific, as per previous publications (Suh, PLOS, 2008; Ugrankar, Dev Cell, 2019).

d. m concerned about drawing strong conclusions from RNAi -KD studies without using Genetic alllesm especially when they are readily available. Even if the authors use multiple independent RNAi's for Act5c, it would be important to use Act5c mutant alleles to corroborate their findings. I appreciate that the Act5c alleles are embryonic lethal. But these alleles are available on the Kyoto stock center as recombined on an FRT (FRT19A) [111764-Act5CG0025 P{ry+t7.2=neoFRT}19A / FM7c; P{ey-FLP.N}5]. Hence, the authors should consider making clones using hsFLP or leaky fat body flippase drivers (see Figure 5; PMID: 19562034 or Figure 4; PMID: 29694367) to make clones of Act5c and then assess- what happens to pLD, mLD; hGLut4 trafficking (Figure 4K) etc.,. This is especially critical for datasets where they over-express hGLut4 (4K) but then KD using RNAi Act5c and conclude that Act5C does not affect trafficking of hGlut4. It is possible that by overexpressing hGlut4 GFP in a hypomorphic condition (RNAi KD), the effect of Act5c in such events is masked. Thus, by using clonal analysis of existing FRT recombined Act5C alleles (from Wagner, 2002- available in stock centers Kyoto), the authors could bolster their conclusions.

Thank you for this suggestion. We attempted to address this issue in several ways. As mentioned, global loss of Act5C is embryonic lethal, so we took advantage of other genetic tools distinct from UAS-RNAi. As suggested we obtained the FRT/FLP Act5C line. This data is now presented in the revised manuscript (Figure 2, Figure Suppl 2). As already noted, we also included two new Act5C RNAi lines we received from VDRC. Strikingly, the FRT/FLP line revealed that in single FB cells, loss of Act5C was sufficient to reduce LD stores and perturb the pLD layer as well as the cytoskeletal ruffling, even when neighboring FB cells with Act5C were fine. This strongly suggests that Act5C loss specifically is required for FB cell LD homeostasis in a cell autonomous manner independent of global effects.

Regarding the hGlut4-GFP, we have not utilized this line in any functional way in the study. We merely express this to show that hGlut4-GFP is still able to traffic to the FB cell surface. However, we have noted during these experiments that the Act5C-depleted animals expressing this hGlut4-GFP still displayed defects in development. We also anecdotally noted that over-expressing hGlut4-GFP did not rescue/suppress the Act5C loss in the FB. We have adjusted the text in the section where we discuss hGlut4-GFP to ensure we only comment on hGlut4-GFP subcellular localization, and no functional aspects in this background.

2. The data on fat-gut lipophorin trafficking is intriguing. Nonetheless, the authors would need to ensure that the Dcg-Gal4 is not expressed in the gut and is fat specific (see point 1c). Furthermore, they need to clarify whether indeed Apolpp secretion is affected. The images of just the Apolpp staining Figure 5E and 5F (TEM) are insufficient evidence to draw a strong conclusion. They can do this by performing western blot with the Apolpp antibodies on hemolymph from the fat-specific Act5CRNAi, as has been shown in Palm 2012 (PMID: 22844248).

Thank you for this point. First, as mentioned above, we have expressed UAS-GFP on Dcg-Gal4 and imaged the larval gut. We find no GFP signal, indicating that Dcg-Gal4 does not express in the gut (Figure Suppl 1). Second, we further investigated how Act5C loss impacted Lpp secretion. Since the electron dense particles we observe in TEM could also potentially be glycogen particles particles, we imaged Glycogenin-YFP in the FB, which labels glycogen particles. We find Glycogenin-YFP has a very distinct localization pattern from what we observe for Lpp in TEM, and this is not significantly altered in Act5C RNAi conditions (Figure Suppl 6).

Second, as requested we have now immuno-blotted for Lpp protein component ApoLII in the hemolymph of control and RNAi-depleted lines (Figure 9). We find that loss of Act5C or Tango1 cause similar reduction of ApoLII signal in the hemolymph, consistent with a FB secretion defect (Figure 9). Consistent with this, we see an accumulation of anti:ApoLII signal in FBs depleted of either Tango1 or Act5C (Figure 9). We do not see this defect in β-spectrin RNAi depletion from FBs, also consistent with our finding that loss of the spectrin network does not impact inter-organ trafficking.

3. Overall, the phenotypes that the author show for Dcg-Gal4>UAS Act5c RNAi- reduced cell size, reduced lipolysis and growth defects. These are all consistent with a defect in insulin signaling. Previous studies have shown that defective Apolpp signaling leads to defective insulin signaling (PMID: 25275323). To increase impact and strengthen the connection between Act5c Apolpp, it would benefit the study, if the authors test whether there is indeed reduced insulin activity in the Act5CRNAi. They can do this by testing p-Akt levels, and tGPH reporter activity.

This is a great point, which we further investigated, and present in our new Figure 6. We find that Act5C loss in the FB reduces insulin signaling (evaluated via qPCR profiles, fluorescent tGPH reporter activity, and reduced DILP2 in the hemolymph). In line with this, we also find that Act5C FB-deficient larvae feed more slowly than control animals.

We also note that ApoLpp-RNAi reduces FB cells size, similar to Act5C loss and lower insulin signaling (Figure Suppl 9). Third, we have now directly assayed Dilp2 levels in the larval hemolymph, and find them significantly reduced in FB-specific Act5C depletion (Figure 6). Collectively, this suggests that Act5C loss influences insulin signaling and feeding, potentially through defective Lpp trafficking. A full dissection of this is beyond the scope of the study here, but we have added discussion of this to our revised manuscript.

4. The authors have investigated Apolpp, but ApoLTP is another transporter specifically between the fat and gut (Palm et al., 2012; PMID: 22844248). Hence, the authors are encouraged to examine ApoLTP localization, and perform experiments with Apoltp.

This is a great point. We have RNAi depleted ApoLTP using a TRiP UAS-RNAi line in other ongoing studies in our group. As expected, ApoLTP loss leads to a very similar depletion of the peripheral LDs (pLDs) in the FB, which we would predict from loss of FB:gut inter-organ trafficking (Ugrankar, Dev Cell, 2019). This is part of another related study in the lab, so we have omitted this observation in this study.

Presentation:1. The authors show that Tango phenocopies Act5C. That is in line with their observation that Apolpp is affected. Therefore it is puzzling that the authors do not cite or discuss the study in 2016 on the role of Tango in ApoB secretion (PMID: 27138255).

Thank you pointing this out. We have now referenced this study.

Similarly, the authors show GRASP puncta and say they affect Golgi secretion. But GRASP could be affecting non-conventional secretion of lipophorins, since it has been shown to affect non-conventional secretion in *Drosophila* (PMID: 18267086 and 29017032). Similarly, the authors should discuss the known role of lipophorin signaling on insulin signaling (PMID: 25275323). From a presentation point of view, the manuscript would benefit from contextualizing observations on secretion and lipophorins with prior studies.

Thank you for this excellent point. We have now further discussed lipophorin and its relationship to insulin signaling in the revision, particularly near the end of the manuscript. Indeed, our observations collectively support a model where defects in lipoprotein trafficking and insulin/nutrient signaling tightly correlate.

2. The current presentation, makes the final conclusions diffuse. Specifically, the last two figures characterize Act5c KD effects on larval development. The manuscript would be more focused if Figure 6 can be moved to supplement and Figure 7 be moved after Figure 2, and the portions with Apolpp RNAi can be presented in an Apolpp specific figure.

We agree that a significant reorganization would enhance the study and its clarity. We have now substantially reorganized several figures to attempt to enhance clarity and narrative flow. The study is now significantly larger (10 Figures, 4 Supplemental Figures). We believe the revised version is more cohesive and provides a more wholistic understanding of FB-specific Act5C depletion.